# Breaking up classroom sitting time with cognitively engaging physical activity: Behavioural and brain responses

Emiliano Mazzoli[1]*, Jo Salmon[1], Wei-Peng Teo[1,2], Caterina Pesce[3], Jason He[4], Tal Dotan Ben-Soussan[5], Lisa Michele Barnett[6]

**1** Institute for Physical Activity and Nutrition (IPAN), School of Exercise and Nutrition Sciences, Faculty of Health, Deakin University, Geelong, VIC, Australia, **2** Physical Education and Sports Science (PESS) Academic Group, National Institute of Education (NIE), Nanyang Technological University, Singapore, Singapore, **3** Department of Movement, Human and Health Sciences, Università degli Studi di Roma 'Foro Italico', Rome, Italy, **4** Department of Forensic and Neurodevelopmental Sciences, Sackler Institute for Translational Neurodevelopment, Institute of Psychiatry, Psychology, and Neuroscience, King's College London, London, United Kingdom, **5** Research Institute for Neuroscience, Education and Didactics, Cognitive Neurophysiology Laboratory, Patrizio Paoletti Foundation, Assisi, Italy, **6** Institute for Physical Activity and Nutrition (IPAN), School of Health and Social Development, Faculty of Health, Deakin University, Geelong, VIC, Australia

* e.mazzoli@deakin.edu.au

**Data Availability Statement:** Since participants have not provided consent for their data to be used for purposes other than those described in the original study aims, the datasets used for this

## Abstract

### Introduction

Classroom-based active breaks are a feasible and effective way to reduce and break up sitting time, and to potentially benefit physical health in school children. However, the effect of active breaks on children's cognitive functions and brain activity remains unclear.

### Objective

We investigated the impact of an active break intervention on typically developing children's cognitive functions and brain activity, sitting/standing/stepping, on-task behaviour, and enjoyment.

### Methods

Up to 141 children, aged between 6 and 8 years (46% girls), were included, although about half of them completed two of the assessments (n = 77, working memory; n = 67, dorsolateral prefrontal cortex haemodynamic response). Classrooms from two consenting schools were randomly allocated to a six-week simple or cognitively engaging active break intervention. Classrooms from another school acted as a control group. The main analyses used linear mixed models, clustered at the class level and adjusted for sex and age, to investigate the effects of the interventions on response inhibition, lapses of attention, working memory, event-related brain haemodynamic response (dorsolateral prefrontal cortex). The mediating effects of sitting/standing/stepping on cognition/brain activity were also explored. To test intervention fidelity, we investigated differences by group on the change values in children's sitting, standing, and moving patterns during class/school time using linear mixed models.

study cannot be made publicly available. However, de-identified versions of the datasets used for the current study may be made available on reasonable request upon approval by the Deakin University Human Research Ethics Committee (DUHREC), Human Research Ethics Office, Deakin University, 221 Burwood Highway, Burwood Victoria 3125, Telephone: +61 3 9251 7129, research-ethics@deakin.edu.au (please quote project number 2016-382).

**Funding:** The study was funded through the Department of Education and Training, State Government of Victoria, Australia (www.education.vic.gov.au). The funders had no role in study design, data collection and analysis, decision to publish, or preparation of the manuscript.

**Competing interests:** JS declares that she has a potential conflict of interest as her husband established a business to manufacture height-adjustable desks for schools in 2017. However, height-adjustable desks were not used in this study and she was not involved in the data analysis. This does not alter our adherence to PLOS ONE policies on sharing data and materials. The other authors declared no competing interests."

Generalized linear mixed models clustered at the individual level were used to examine on-task behaviour data. For the intervention groups only, we also assessed children's perceived enjoyment, physical exertion and mental exertion related to the active breaks and compared the results using independent t-tests.

## Results

There was a significantly greater positive change in the proportion of deoxygenated haemoglobin in the left dorsolateral prefrontal cortex of children assigned to cognitively engaging active breaks compared to the control group ($B = 1.53 \times 10^{-07}$, 95% CI [$0.17 \times 10^{-07}$, $2.90 \times 10^{-07}$]), which under the same cognitive performance is suggestive of improved neural efficiency. Mixed models showed no significant effects on response inhibition, lapses of attention, working memory. The mediation analysis revealed that the active breaks positively affected response inhibition via a change in sitting and standing time. The sitting, standing, and moving patterns and on-task behaviour were positively affected by the active breaks at end of trial, but not at mid-trial. Children in both intervention groups showed similarly high levels of enjoyment of active breaks.

## Conclusion

Cognitively engaging active breaks may improve brain efficiency in the dorsolateral prefrontal cortex, the neural substrate of executive functions, as well as response inhibition, via effects partially mediated by the change in sitting/stepping time. Active breaks can effectively reduce sitting and increase standing/stepping and improve on-task behaviour, but the regular implementation of these activities might require time for teachers to become familiar with. Further research is needed to confirm what type of active break best facilitates cognition.

## Introduction

Physical activity can benefit children's physical, social and cognitive health [1,2]. Executive functions are cognitive functions associated with core and higher-order thinking processes, and behaviour regulation [3], and are crucial for success in personal, social, academic and professional activities [3]. Three core executive functions have been identified [4]: inhibition, updating (or working memory), and shifting (or cognitive flexibility). A meta-analysis [5] concluded that physical activity interventions, especially those increasing moderate-to-vigorous intensity duration, benefited non-executive (effect size [ES] = 0.23), executive (ES = 0.20) and metacognitive functioning (ES = 0.23), albeit with small effects. In 2019, Singh et al. [6] concluded that the effects of physical activity on cognition still appear inconclusive. In contrast, Tomporowski and Pesce [7] have argued that the relationship between physical activity and cognitive functions is more consistent when the cognitive engagement needed to perform motor skills in physical activity is taken into account (individually or interactively with the metabolic exercise demands).

The hypothesised mechanisms for effects of physical activity on cognition are mostly physiological in nature; for example, the increased cerebral blood flow as a consequence of increased physical activity [8]. The inherent cognitive stimulation of a physical task that is also mentally engaging might produce greater effects on cognition than simple physical activity [9], perhaps

because the recruitment of certain brain regions supports neuroplasticity (i.e., the adapting ability of the brain in response to the changing demands) [10]. At the cortical level, executive functions have been associated with cerebral activity in the dorsolateral prefrontal cortex (DL-PFC) [11], the same brain region activated by cognitively demanding physical activity [12]. Thus, the neural engagement of specific brain networks may enhance the neural efficiency of those networks [13], and facilitate the mental processes associated with them—i.e., executive functions. Neural/brain networks are expensive from both structural (wiring cost) and functional (metabolic cost) perspectives [14]. Neural/brain efficiency can be understood as a cost-effective organisation of the neural connections and management of the metabolic (or energetic) resources, in a trade-off aimed at reducing the costs and maximising the capacity to process information [14]. Cognitive functions are traditionally assessed by analysing the behavioural responses to a computer-based test that supposedly challenges certain functions, and the performance is generally quantified in terms of response time and accuracy rate. The concurrent use of objective measures of brain activity might provide complementary evidence of the underlying mechanisms that support changes in cognitive performance, although such approaches have rarely been applied to physical activity interventions. The most advanced method to investigate changes in brain structure and function with high spatial resolution involves the use of neuroimaging techniques, such as functional magnetic resonance imaging. A recent systematic review identified nine studies that employed neuroimaging techniques in youth to test the effects of physical activity interventions, with findings from seven randomised controlled trials included in the review showing significant improvements in brain structure and/or function [15]. Despite these promising findings, most neuroimaging devices are non-portable and high cost, which may explain the paucity of physical activity research using this technique. Another approach is to measure event-related brain potentials using electroencephalography, which—although has high temporal resolution and is significantly cheaper than most neuroimaging techniques—is not easily portable, has low spatial resolution, and is very sensitive to motion artefacts. Functional near-infrared spectroscopy (fNIRS) is a relatively novel optical technique that allows to measure brain changes of oxygenated ($O_2Hb$) and deoxygenated (HHb) haemoglobin (i.e., haemodynamic response) [16]. Its high spatial and temporal resolution, high biochemical specificity, the portability, and the relative stability to motion artefacts make fNIRS greatly advantageous in terms of ecological validity compared to other techniques [17]. A fast-growing number of studies, mostly cross-sectional in design, have employed fNIRS in youth [18]. To the authors' knowledge, only two cross-sectional studies [19,20] and no interventions pertaining to children's physical activity and brain function have been conducted using fNIRS.

Aside from the cognitive and other health benefits from being physically active, sedentary behaviour is considered by some to be a risk factor for health even after adjusting for physical activity [21]. Although the evidence for this relationship appears robust in adults [22], it remains inconsistent in children, particularly for objectively assessed sedentary time and cardiometabolic health outcomes [23]. Research on sedentary behaviour and cognitive functions in children is still at an early stage and the findings reported so far appear inconsistent. For example, Syväoja et al. [24] found that greater sedentary time was associated with better attention amongst 12-year-olds, whereas Mazzoli et al. [20] found that higher objectively measured sitting time was associated with more attentional lapses during the performance of an executive functioning task in 7-year-olds. Lapses of attention can hinder a person's ability to perform everyday tasks [25] and are generally seen as conflicting with learning [26].

Reducing sitting while increasing physical activity during school hours may have potential to effectively contribute to children's physical and cognitive health [27]. The integration of movement in the classroom may take place in different forms [28], including active breaks

(i.e., short bouts of physical activity that can be related or unrelated to the curricular content), active lessons (i.e., curricular activities taught with movement, such as active Maths), and changes in the classroom environment (e.g., the use of adjustable height-adjustable desks). All these strategies are low-cost [29,30] and reduce sedentary behaviour [31]. Active breaks during class time provide the opportunity to break up sitting time and increase physical activity in the school setting without affecting the curricular structure and are very easy to learn and implement by teachers [32].

Children's enjoyment [33] (i.e., the main component of intrinsic motivation [34]), is one of the factors that influence the successful implementation of physical activity programs, including active breaks. Enjoyment is positively related to physical activity [35], is negatively associated with sedentary behaviour [36], and is also hypothesised to positively affect executive functions [37]. Whilst classroom-based physical activity has been previously positively associated with greater levels of children's enjoyment compared to traditional lessons [38], no studies have investigated whether enjoyment levels could be differently affected by cognitively engaging or simple active breaks.

Over the last few years, a number of studies have shown the effectiveness of these strategies in breaking up sitting and producing desirable health-related benefits [39,40]. While most of these studies have focused on the physical outcomes, more recent research has explored the impact of reducing and breaking up sitting on cognitive outcomes. The cognitive constructs most frequently investigated were core executive functions (i.e., inhibition [interference and attention], working memory and cognitive flexibility) [41–45], higher-order executive functions (i.e., fluid intelligence) [46,47], memory [48] and time on-task [49–51]. Although these aspects are relevant to academic performance [52–55], other cognitive functions might be equally important to study. As noted earlier, the cognitive engagement required to perform certain physical activities needs to be considered [7]. However, no studies have explored the effects of active breaks with different cognitive requirements on response inhibition, lapses of attention, and brain activity (i.e., DL-PFC haemodynamic response), all important elements of most academic activities. Furthermore, only four previous studies [45,48,56,57] have investigated the effects of cognitively challenging types of physical activity to break up sitting in the classroom, three of which [45,48,56] found positive effects on cognition/learning using this approach. None of these studies used both behavioural and neural measurements, or objectively assessed children's sitting/standing/stepping patterns.

Therefore, the main aim of this study was to test the effects of a six-week active break intervention with two conditions (simple [low cognitive engagement] and cognitively engaging) on primary school children's response inhibition, lapses of attention, working memory and DL-PFC haemodynamic response, as compared to a control group (normal school practice with no breaks apart from recess and lunch). We additionally explored whether the effects of active breaks on cognitive functions or DL-PFC haemodynamic response could be mediated by the change in class time sitting, standing, or stepping. Furthermore, we aimed to investigate the effects of both interventions on children's sedentary patterns, on-task behaviour, and enjoyment, as secondary outcomes. We additionally tested children's perceptions of the physical and mental effort related to the active breaks to check whether the manipulation of the physical and mental components of the breaks designed for different intervention groups was implemented successfully by researchers.

We hypothesised that: i) response inhibition and working memory would improve more in the active conditions than the control; ii) attentional lapses would decrease in the active conditions more than the control; iii) DL-PFC haemodynamic response changes would reflect improved efficiency in the active conditions compared to the control; and iv) the cognitively engaging intervention would show greater improvements in each measure compared to the

simple intervention. For the secondary analysis, we hypothesised that: i) the intervention groups would show a similar reduction in class time sitting and/or increase in standing/stepping, compared to the control group; ii) changes in sedentary patterns would mediate better cognitive performance and/or more efficient DL-PFC haemodynamic response in the intervention groups, with the cognitively engaging intervention showing greater effects; iii) the odds of observing children's behaviour as on-task would appear significantly reduced during the second observation, compared to the first, in the control group but not in the intervention groups, at mid-trial and end of trial; iv) both active conditions would show similar levels of enjoyment and physical exertion; and v) the cognitively engaging intervention would report that the breaks were more cognitively demanding compared to the control.

## Methods

Data for this controlled trial titled "Active breaks in the classroom to improve thinking skills" was collected from October 2017 to December 2017. The trial was registered in the Australian New Zealand Clinical Trials Registry (registration number ACTRN12618002034213), which includes all ongoing and related trials for this intervention. The trial registration was retrospective as the researchers did not complete this prospectively but thought it would still be useful. A description of the trial protocol is available as a supporting information file.

### Ethics approval and consent

The study received approval by Deakin University Human Research Ethics Committee on the 25[th] of January 2017 (2016–382) and by the Department of Education and Training of Victoria (2016_003257). Based on the results of a previous feasibility study [32], the present trial was conducted with children from grades 1 and 2. Recruitment of participants was carried out between March and October 2017. Schools, teachers, and parents (on behalf of their children) provided informed written consent to be part of this study. Parents/guardians of children participating in the study were invited to complete a demographic survey at the time of consent. This study was performed in accordance with the standards of ethics outlined in the Declaration of Helsinki.

### Participants

To be eligible to participate in the study children had to be i) typically developing ii) aged between 6 and 9 years, and iii) attending Grade 1 and 2 in a mainstream primary school. Exclusion criteria included: i) having a visual or auditory impairment, as most of the primary outcomes were measured with assessments not designed for children with these types of impairments; and ii) having a physical impairment that would not allow children to participate in the breaks.

The sample size was determined based on previous research [e.g., 56] and a well-documented small to moderate effect size of physical activity on executive functions [e.g., 5,58]. We aimed to recruit around 43 children with typical development per study arm ($\sim$N = 130 children) and four teachers per study arm ($\sim$N = 12 teachers). Three primary schools in Melbourne were recruited using convenience sampling, with 153 consenting children aged around seven years, from 15 classrooms. One child changed school and one withdrew before study commencement; another two children withdrew while the study was conducted and eight were absent on at least one of the assessment days. Overall, 141 children had a measure of response inhibition at baseline and end of trial. Working memory and DL-PFC haemodynamic response assessments were limited to a random sub-sample of children that could be completed in the school time available for these assessments. Hence, 77 children completed

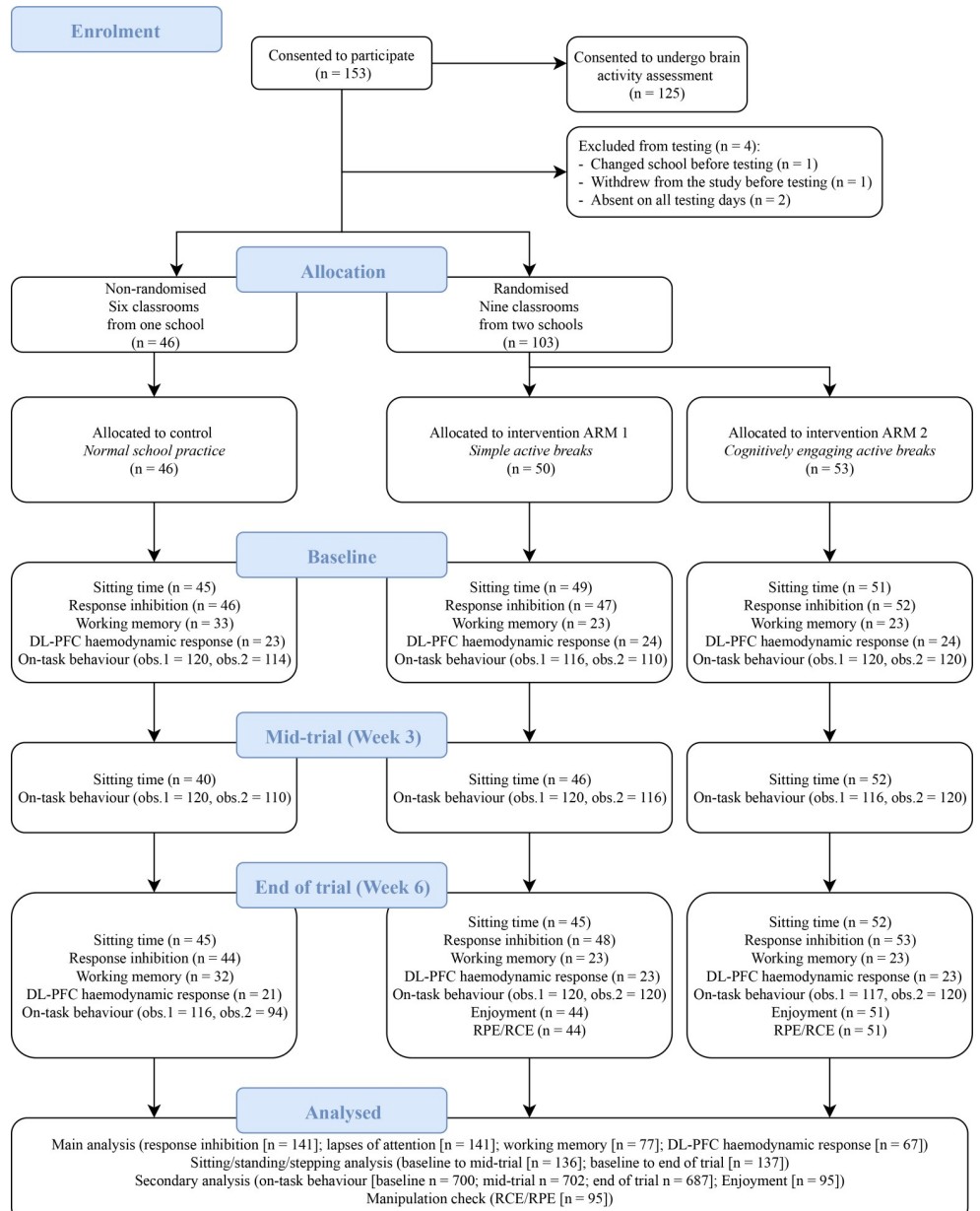

**Fig 1. Recruitment flow diagram according to the Consolidated Standards of Reporting Trials (CONSORT) [59].**
Working memory assessment was limited to a total sub-sample of 79 children randomly allocated, from the total sample of 153 children, due to the limited time available to complete this assessment. Dorsolateral prefrontal cortex (DL-PFC) haemodynamic response assessment was limited to a random sub-sample of 71 children, out of the 125 who provided additional consent to undergo the DL-PFC haemodynamic response assessment, due to the limited time available to complete this assessment. At each time point, on-task behaviour was assessed with six children randomly selected from each group (one classroom per study group); for this assessment, two consecutive 30-min observations were completed at each time point. Rating of perceived physical effort (RPE); rating of perceived cognitive effort (RCE).

working memory and 67 completed the DL-PFC haemodynamic response assessment at baseline and end of trial. Fig 1 shows the recruitment flow diagram according to the Consolidated Standards of Reporting Trials (CONSORT) [59].

## Randomisation

The classrooms of two schools were randomly assigned by EM (i.e., computerised sequence generation with random.org) to cognitively engaging active breaks (physically active with high cognitive engagement; classrooms n = 5) or to simple active breaks (physically active with low cognitive engagement; classrooms n = 4). Another school continued with the normal school practice (control; classrooms n = 6). There were on average 10 consenting children per class. The school allocated to control only agreed to participate as a control group, therefore, we could not randomly allocate this condition.

## Active breaks intervention arms

After the allocation to one of the two intervention arms, teachers in the intervention groups attended a one-off 20-min face-to-face theoretical and practical training session on how to conduct the active breaks in their classrooms. Teachers were asked to select the active breaks from a specific repertoire of seven activities, following a regular rotation, and to use these breaks to interrupt children's prolonged sitting twice a day (between 9:00 am and 11:00 am and between 11:30 am and 1:00 pm) for six weeks. All the activities were designed to last between four and five minutes, but each intervention arm was provided with a different set of active breaks that reflected different levels of cognitive engagement. That is, cognitively engaging active breaks were meant to be more cognitively effortful for participants, compared to simple active breaks.

Teachers were provided with a hard copy of a manual, including a description of the activities, specific instructions to be followed for each session, an activity log to record teacher's daily progress, suggestions on additional resources and equipment that could be used, as well as some equipment (i.e., a light-weight ball, visual cards, action prompts, dice, and music). An example of a simple active break was the game *Quick fit*!, a simple imitation of a movement sequence. The cognitively engaging counterpart was the game *My Clock is Late*!, an imitation of a coordination sequence with a time delay between teacher and children (i.e., similar to singing in rounds, but with specific activities instead) [60]. The activities provided to teachers in the intervention groups are summarised in S1 Table. Teachers in the control group were not involved in any training sessions and were asked to continue with usual school activities. The trial was carried out for six weeks, between October and December 2017.

## Measures and data management

The following sections present an overview of the other measures used in the study; further details on the measures and data management are available elsewhere [20]. Data on children's cognitive functions and DL-PFC haemodynamic response was collected at baseline and at the end of the trial, on school days not involving an assessment of children's sedentary behaviour. This was conducted in a quiet room within the school premises. On-task behaviour was measured via systematic classroom observations at baseline, mid-trial and at the end of the trial.

**Demographic information.** Participants' characteristics, including children's date of birth, language spoken at home, parental background, education, occupation, and income, were collected using a parent survey.

**Primary outcomes.** *Response inhibition and lapses of attention*. A Go/No-Go task [61] was used to measure response inhibition and attentional lapses. The Go/No-go task used in the present study was a computer-based task paradigm, programmed in E-prime 2.0 (Psychology Software Tools, Pittsburgh, PA, USA), which presented participants with a pseudo-random series of white (Go trials) or yellow (No-go trials) circles. Participants were instructed to press the space bar of the laptop keyboard on Go-trials and to withhold their responses on

No-go trials. It was emphasised that participants should try to respond as quickly and as accurately as possible. Each child's task performance was summarised as an Inverse Efficiency Score (IES) [62], which combines the average response time on Go-trials (i.e., when they pressed the spacebar upon stimulus presentation) and the proportion of No-go trials in which their response correctly withheld. IES is obtained by dividing response time by the proportion of accurate responses; a lower IES is indicative of an overall better performance at the test.

Lapses of attention (i.e., temporary failure of goal-directed behaviour) are momentary distractions known to interfere with the performance of tasks that require focus. When participants perform a task that requires responding to a certain set of stimuli, lapses of attention are commonly identified as an occasional, but exceptionally long response time. These can be calculated by fitting exponential-Gaussian distributions to an individual's response time distributions and extracting the exponential component ($\tau$) [63]. For each participant, we extracted $\tau$ using R Statistical Software (Version 3.5.1, The R Foundation for Statistical Computing, Vienna, Austria), R Studio (Version 1.1.463., RStudio Inc., Boston, MA, USA), and the package "retimes" [64].

*Working memory*. Children's working memory was measured using the National Institutes of Health (NIH) Toolbox List Sorting Working Memory Test, an iPad-based cognitive test which demonstrated excellent test-retest reliability and adequate convergent and discriminant validity in children and adolescents (3–15 years) [65]. The test requires children to recall a set of animals or foods that are presented in each trial and to list them in size order from the smallest to the largest. The test results are reported as a raw score (range 0–26), unadjusted standardised score (derived comparing each participant's result to a normative sample), and a standardised score adjusted for age. The unadjusted standardised score was used for the analyses.

*Event-related haemodynamic response in DL-PFC*. Children's left DL-PFC haemodynamic response was measured using a portable single-channel continuous-wave functional near-infrared spectroscopy (fNIRS; PortaLite, Artinis Medical Systems, The Netherlands). Given the relative transparency of human tissues to near-infrared light, the emission of near-infrared light into a tissue and the measurement of the intensity of re-emerging light enables assessment of changes in $O_2Hb$ and $HHb$ non-invasively in that specific region [17]. Specific details on the technological and methodological aspects related to fNIRS is available several previous reviews [e.g., 66]. Each child involved in this assessment had the fNIRS probe fitted by the corresponding author (EM) on their forehead in the area corresponding to the left DL-PFC. From the frontal aspect of forehead, the landmark corresponding to the left DL-PFC was identified as the area between the mid-point of the left eyebrow and the hair growth. The 10–10 international system [67] was used to position the probe's light source and detector (in AP3 and F5, respectively), in line with the approach described by Zimeo Morais et al. [68]. The probe was secured in place using a dark elastic fabric hairband. The probe size is $58 \times 28 \times 6$ mm, and it fits quite precisely on the small forehead of a child. No skin preparation was required but the researcher ensured that no hair was in the way and that the incoming signal was good prior to starting the assessment. An illustration of the probe and the experimental setup is available in Fig 2.

For each child, the real-time changes in $O_2Hb$ and $HHb$ in the left DL-PFC were recorded while performing a 10-min task involving inhibition abilities (Go/No-Go task). Collected data were processed, and artefacts were removed, using the hemodynamic optically measured evoked response (HomER2), a MATLAB-based (MATLAB R2017a, MathWorks, Natick, MA, USA) user interface developed by Huppert and colleagues [69] and artefacts were removed as described by Brigadoi et al. [70]. The artefacts removal process involved the following steps: i) raw data were converted in optical density changes; ii) a filtering algorithm was applied to

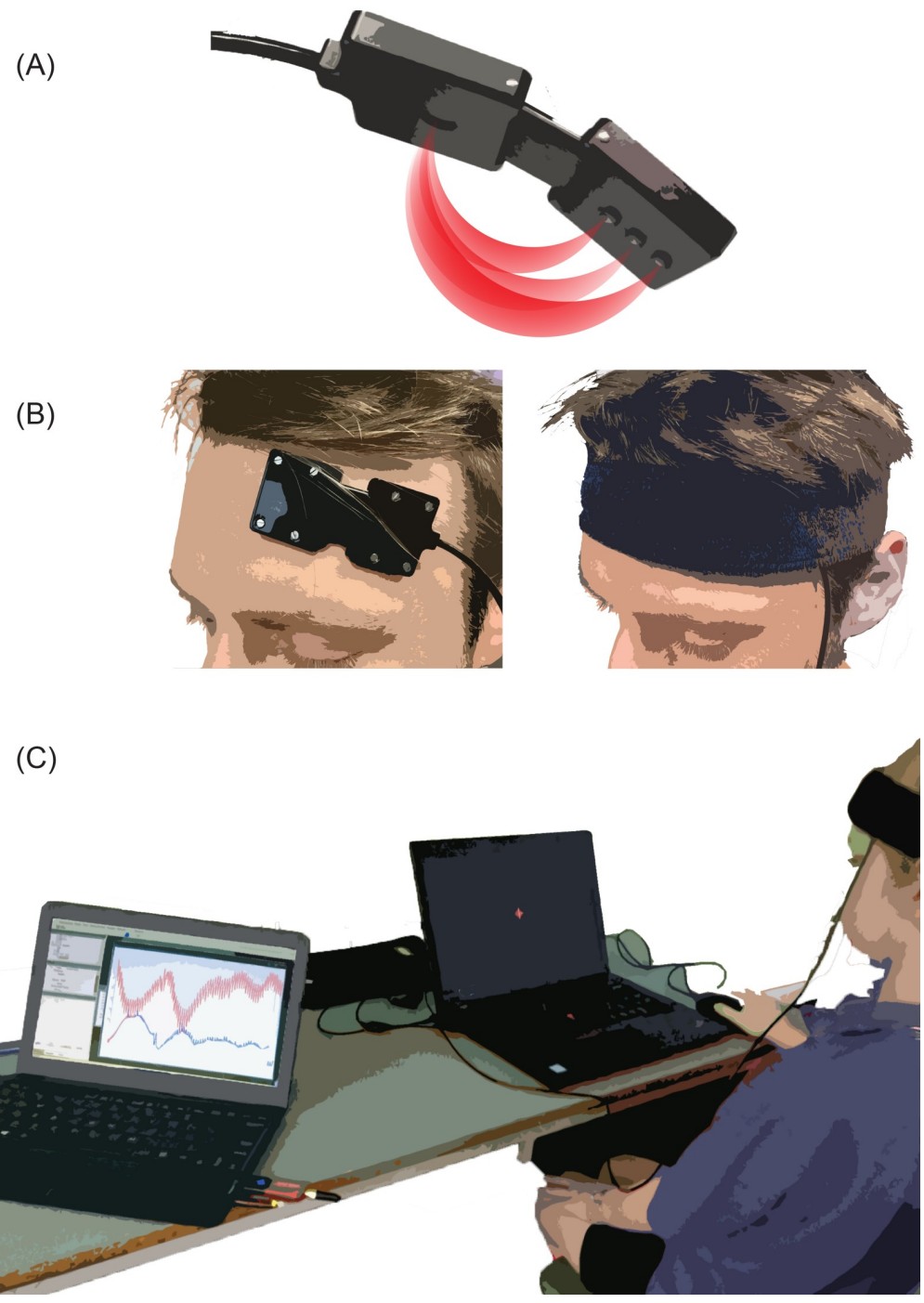

**Fig 2. Functional Near-Infrared Spectroscopy (fNIRS) device and experimental setup.** (**A**) The Portalite fNIRS probe (Artinis Medical Systems) with graphical representation of the three light sources (right) and the detector (left); (**B**) placement of the probe; (**C**) experimental setup.

detect motion artefacts; iii) a principal component analysis (PCA) was applied to correct the detected motion artefacts; iv) an high pass filter (0.010 Hz) and a low pass filter (0.20 Hz) were applied to clear the data from the high and low frequency noise; and v) optical density data

were converted in concentration changes and used for the analysis. A detailed description of the utilised pipeline values for data processing, also including the MATLAB script, is available as supporting information (S2 Table). The average changes in $O_2Hb$ and HHb, during Go and No-Go blocks, were calculated and used for data analysis. These values were extracted between 10 and 30 seconds of each test block, to account for the delay in the haemodynamic response time [71].

**Secondary outcomes.** *Sitting, standing, stepping patterns.* Children were fitted with an *activ*PAL™ accelerometer for two school days at baseline, mid-trial and at the end of the trial, to capture their sitting, standing, and stepping patterns on typical school days. Assessment days excluded physical education or school sport timetabling, to avoid the confounding effect of those activities. Two researchers fitted the monitors on the child's front thigh, using an adjustable elastic band, at the beginning of the school day (before 9:30 am) and instructed the children to keep them on their thigh throughout the school day and to return the monitors to the teachers at the end of the school day (3:30 pm). Researchers collected the monitors at the end of each school day and downloaded the data at the end of each second day. The average daily sitting, standing, and stepping time, sit-to-stand transitions, frequency of and time spent in sitting bouts greater than 5 min and greater than 20 min, and total step count during school time and class time were calculated using a previously designed MS® Excel Macro (Microsoft Corporation, Redmond, WA, USA) and STATA 15.0 (Stata Statistical Software, Release 15, StataCorp LLC, College Station, TX, USA). Class/school data collected for at least one of the two school days of measurement at each time point was identified as valid using a 50% wear time criterion [72]—consecutive zeros for 20 min or more were identified as non-wear time [73]. To allow the comparison between children with different wear times, each variable was standardised according to wear time.

*On-task behaviour.* Systematic observations of children's on-task behaviour were conducted by two trained observers in one classroom per intervention/control group, selected at random at each time point (baseline, mid-trial, and end of trial). This required a researcher to sit quietly in a corner of the classroom for an hour and to observe six consenting children (selected at random) following the prompts coming from a previously recorded audio file. Each child was observed for 10 seconds, after which the observed behaviour was noted down (5 seconds). After four consecutive observation intervals the next child was observed. This method has been previously used successfully in the classroom [74]. An inter-observer reliability between the two observers was calculated following the method suggested by Mahar [74], revealing an overall percentage of agreement of 91.1%.

Each observable child's behaviour was described and grouped into four categories hereafter summarised: i) on-task behaviour (the child's focus is on the task assigned by the teacher); ii) off-task noise (the child is not focused on the work assigned by the teacher, he/she is talking, etc.); iii) off-task motor (the child is not focused on the work assigned by the teacher, he/she is walking around, etc.); or iv) off-task other (the child is not focused on the work assigned by the teacher, his/her behaviour is a combination of 2 and 3 or something different). For an interval to be scored as on-task, the behaviour being observed should have persisted for the entire interval (i.e., whole interval recording). Instead, in the case of off-task behaviour, we adopted the partial interval recording, where the behaviour is scored as off-task if it occurs at all during the interval. Whilst this behaviour is recorded as per instrument protocol, in the analysis we treated on-task behaviour as a dichotomous variable (i.e., on-task/off-task).

Data were collected at each time point with one randomly selected classroom per group condition (six consenting children also selected at random in each classroom) for two consecutive 30-min periods of academic instruction. The academic instruction was uninterrupted for all groups at baseline. At mid-trial and end of trial the intervention groups had an active

break in between the two observation periods, the control continued with the uninterrupted academic practice.

*Children's enjoyment and physical/mental effort.* At end of trial, children who were in the intervention arms completed a modified version of the Physical Activity Enjoyment Survey (PACES) [75], designed to understand their enjoyment related to the active breaks. The PACES was modified by replacing the first sentence (i.e., "When I am active. . .") with "When I do active breaks in the classroom. . .", to allow children to direct their answers towards the active breaks instead of general physical activity. To test whether the manipulation of physical and mental components of the active breaks reflected the researcher's assumptions: (i) children's perceptions regarding the physical effort required to participate in the active breaks was assessed using a pictorial scale for physical exertion valid for use in children [76]; (ii) using a similar approach, a pictorial scale to measure mental effort was used by researchers to test the successful manipulation of cognitive engagement in the two active conditions, as previously done by Schmidt et al. [45].

*Intervention fidelity.* The differences by study group on children's sitting, standing, and moving patterns over time during class/school periods were used to assess teacher's adherence to the program. Additional information was retrieved from teachers' activity logs, which were designed to allow teachers to record the number and type of active breaks performed on each trial day.

## Statistical analysis

**Preliminary analysis.** Summary statistics of the sample demographic information, sitting pattern, cognitive performance, and DL-PFC haemodynamic response were calculated. Differences by study group in demographic characteristics (i.e., age, sex, reported medical/developmental condition, primary language spoken at home) were calculated using Analysis of Variance (ANOVA) or $\chi^2$ test according to the nature of the data. The outcome variables of the main analysis (all continuous) were assessed for normality with histograms, Q–Q plots and by examining skewness and kurtosis values. A visual inspection of the haemodynamic response and the concurrent change in cognitive performance was conducted prior to further analysis.

**Main analysis.** Separate linear mixed models were conducted to investigate the effects of being part of a study group on each cognitive or DL-PFC haemodynamic response outcome. For each model, the relative change between baseline and end of trial was used as an outcome measure. All models were adjusted for sex and age—commonly identified as potential factors affecting cognitive functions, physical activity, and sedentary behaviour—and the baseline value of the outcome variable—to avoid regression to the mean [77]—and accounted for the random effects of classroom as a clustering variable. The models examining DL-PFC haemodynamic response were also adjusted for the performance change score in the cognitive task, to account for the variability in haemodynamic response that could be explained by a change in cognitive performance. A measure of effect size (Cohen's $f^2$) was provided for fixed effects of the study groups, which was calculated according to the method described by Selya et al. [78]. This operation requires dividing the proportion of variance explained by the predictor of interest by the residual variance not explained by the model. Conventionally, the effects are considered small for $f^2 = 0.02$, moderate for $f^2 = 0.15$ and large for $f^2 = 0.35$ [79]. Based on the observed effect size for each of the main outcomes, the actual sample size, and an $\alpha$ error probability = 0.05, a post-hoc power analysis was conducted to determine the attained power ($1 - \beta$ error probability), also considering the design effect correction required to account for the random effects of class as a clustering variable. The design effect was calculated using the

formula: $1 + [(CV2 + 1) \times n- 1] \times (ICC)$, where CV indicates the coefficient of variation for n; n is the number of students in each classroom (cluster); and ICC is the intraclass correlation coefficient from the linear mixed models.

**Secondary analysis.** To test fidelity of the intervention, linear mixed models examined the effects of the active breaks on the change values in sitting, standing, and stepping patterns across class time and school time. Additionally, teachers' activity logs were inspected.

Following the method for mediation outlined by VanderWeele [80], we tested whether the effects of the intervention on cognition/DL-PFC haemodynamic response could be mediated by the change in sitting, standing, and stepping time. We fitted one linear regression for each outcome and one for each mediator, and used the coefficients resulting from each of these models to calculate the direct, indirect, and total effects of each mediator on each outcome variable. We also tested the existence of an interaction between exposure and mediator, to see if the effects of the exposure on the outcome differed at different levels of the mediators. We tested each intervention condition (cognitively engaging/simple active breaks) against the control and adjusted all models for children's sex and age; DL-PFC haemodynamic response was also adjusted for the change in cognitive performance over time. The analyses were conducted using STATA® 15.0 and the module PARAMED [81]. One of the advantages of this using PARA-MED is that it allows to test the effects of interaction terms in the mediation model. As suggested by Hayes [82], we used bootstrapping (based on 1,000-sample bootstraps) to calculate bias-corrected 95% confidence intervals (CI) of the estimated direct, indirect, and total effects.

On-task behaviour data was summarised descriptively across the observation periods by group. We used multilevel generalized linear models to investigate the difference for children in the intervention groups in the odds of being observed on-task in the two consecutive 30-min periods compared to those in the control group, at each study time point. Post-trial data on children's enjoyment was presented descriptively and analysed using independent-samples t-test, to investigate possible differences in children's enjoyment of the two different interventions. To investigate whether researchers successfully manipulated the physical and cognitive components of the active breaks independent t-tests were also used to examine differences between the active break conditions on children's perceptions of the physical and cognitive effort in relation to the active breaks. All the analyses were conducted using STATA 15.0.

## Results

### Demographic information

A maximum of 141 children aged between 6 and 8 years (46.1% girls) were included in the main analyses. The sample size varied for the different outcomes. A total of 15 children were reported to have a medical or developmental condition including asthma (n = 3), sensory processing disorder (n = 2), dyslexia (n = 2), attention-deficit/hyperactivity disorder (n = 2), autism spectrum disorder (n = 2), speech delay (n = 1), hypercalciuria (n = 1) and global developmental delay (n = 1); one condition was not specified by parents/guardians. Since such conditions may have modified the effects on the main outcome, the results of the main analysis without those children have also been reported. No participants were excluded based on their adherence to the program. Children's summary statistics are presented in Table 1 and parental characteristics are in Table 2.

### Main analysis

Outcome variables appeared approximately normally distributed. A preliminary inspection of the DL-PFC haemodynamic response and the related cognitive performance suggested that

**Table 1. Demographic characteristics of children included in the main analysis.**

| Children's characteristics | Control | Simple | Cognitively engaging | Total | P-value |
|---|---|---|---|---|---|
| N | 43 | 46 | 52 | 141 | |
| Age in years at baseline, M (SD) | 7.7 (0.6) | 7.7 (0.6) | 7.6 (0.6) | 7.7 (0.6) | 0.82 |
| % Girls | 44.2 | 41.3 | 51.9 | 46.1 | 0.55 |
| % Medical/developmental condition | 11.6 | 15.6 | 5.8 | 10.7 | 0.29 |
| % Primary language: English | 97.7 | 87.0 | 90.4 | 91.5 | 0.18 |

$P$-values were calculated using Analysis of Variance (ANOVA) or $\chi^2$ test according to the nature of the data.

both intervention groups showed lower cerebral activity and improved response time at the cognitive task compared to the control, by the end of trial (Fig 3).

The results of the main analysis are displayed in Fig 4.

By the end of trial, children in the cognitively engaging intervention showed a significant reduction in the response time at the response inhibition task (approximately –15.40 ms), while maintaining accuracy at the same level (Table 3). Additionally, children in this group showed a significant reduction in attentional lapses (approximately –7.97 ms). Neither response inhibition nor lapses of attention improved significantly by the end of trial for

**Table 2. Demographic characteristics of parents.**

| Socio-economic characteristics | Control group | | Simple intervention | | Cognitively engaging intervention | | Total | |
|---|---|---|---|---|---|---|---|---|
| | P1 | P2 | P1 | P2 | P1 | P2 | P1 | P2 |
| n | 43 | 43 | 45 | 45 | 49 | 52 | 137 | 140 |
| Country of origin | | | | | | | | |
| % Australia | 67.4 | 76.2 | 68.9 | 70.5 | 63.3 | 71.4 | 63.3 | 72.6 |
| % Other | 32.6 | 23.8 | 31.1 | 29.5 | 36.7 | 28.6 | 36.7 | 27.4 |
| Parental education | | | | | | | | |
| % University or tertiary education | 90.7 | 97.7 | 70.5 | 80.0 | 77.6 | 76.4 | 79.4 | 84.3 |
| % Other [a] | 9.3 | 2.3 | 29.5 | 20.0 | 22.4 | 23.1 | 20.6 | 15.7 |
| Parental employment status [b] | | | | | | | | |
| % Employed full time | 76.7 | 34.9 | 84.1 | 22.2 | 87.8 | 21.2 | 83.1 | 25.7 |
| % Employed part time | 11.6 | 58.1 | 6.8 | 48.9 | 8.2 | 46.2 | 8.8 | 50.7 |
| % Other [c] | 11.6 | 14.0 | 11.4 | 31.1 | 6.1 | 34.6 | 8.8 | 26.4 |
| **Parental combined income** | | | | | | | | |
| n | 40 | | 40 | | 49 | | 129 | |
| % < AUD 30,000 | - | | 2.5 | | 6.1 | | 3.1 | |
| % AUD 30,000–59,000 | 5.0 | | 7.5 | | 14.3 | | 9.3 | |
| % AUD 60,000–119,000 | 10.0 | | 15.0 | | 14.3 | | 13.2 | |
| % AUD 120,000–180,000 | 37.5 | | 40.0 | | 34.7 | | 37.2 | |
| % > AUD 180,000 | 47.5 | | 35.0 | | 30.6 | | 37.2 | |

P1, Father / guardian; P2, Mother / guardian.

[a] Primary school; some high school; completed high school; technical/trade certificate/apprenticeship; not applicable.

[b] The number of responses may exceed the number of respondents due to multiple responses allowance for this field.

[c] Home-duties full time; unemployed/unpaid; student; not applicable. Percentages may not sum up to 100 due to rounding. The total number of responses for each characteristic might not equal the total number of responses because missing responses have not been reported.

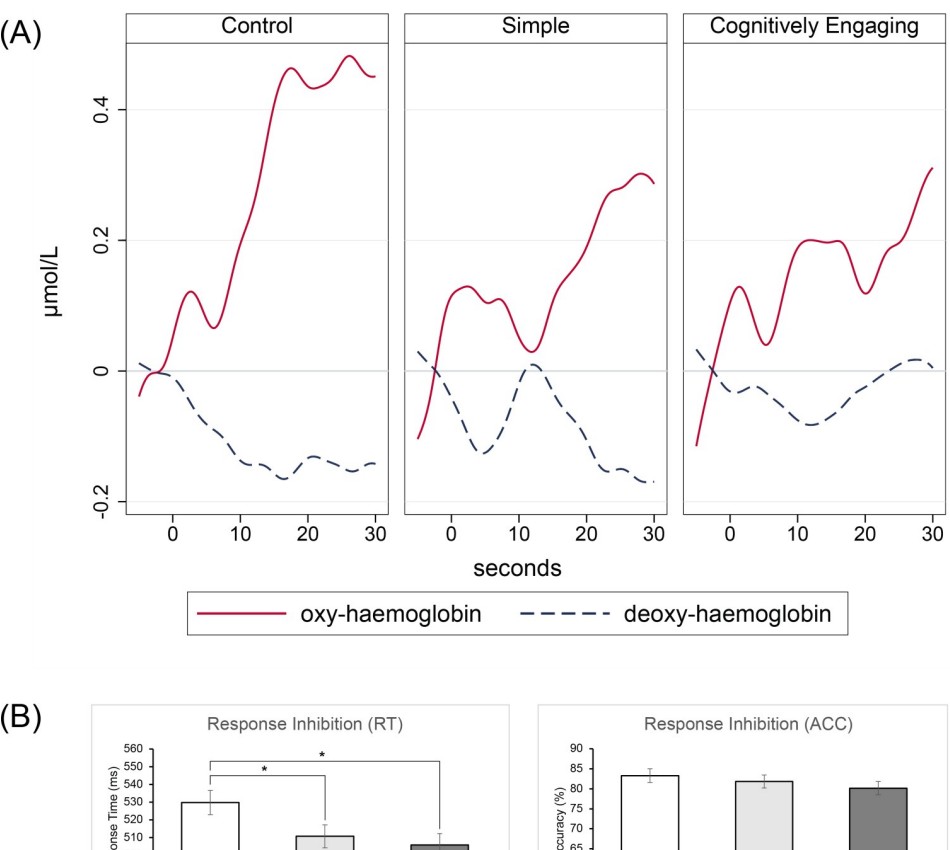

**Fig 3. Left dorsolateral prefrontal cortex haemodynamic response and related cognitive performance at the end of trial.** (**A**) Unadjusted average haemodynamic response (oxy- and deoxy-) during a Go/No-Go task at the end of trial by study group; (**B**) Performance at the Go/No-go task completed during the fNIRS assessment; group marginal means ± standard error calculated following linear mixed models adjusted for the baseline performance at the task, sex, age, and for the random effects of classroom as a clustering variable. Note that lower response time (RT) and/or higher accuracy (ACC) are indicative of better performance at the cognitive test.

children in the simple intervention or the control group. Working memory significantly improved in all groups by the end of trial, compared to baseline. However, neither intervention significantly affected children's working memory, compared to control. For children's DL-PFC haemodynamic response, from baseline to end of trial the control group showed a significant increase in $O_2Hb$ (approximately $+3 \times 10^{-07}$ mol/L) and decrease in HHb (approximately $-1 \times 10^{-07}$ mol/L), while maintaining the cognitive performance at the same level. Compared to the control, the cognitively engaging group showed a significantly greater positive change in HHb compared to the control ($B = 1.53 \times 10^{-07}$ mol/L, 95% CI [$0.17 \times 10^{-07}$, $2.90 \times 10^{-07}$], $p = 0.028$). On the other hand, the effects of the simple intervention on $O_2Hb$ approached significance ($B = -3.20 \times 10^{-07}$ mol/L, 95% CI [$-6.54 \times 10^{-07}$, $0.15 \times 10^{-07}$], $p = 0.061$; simple intervention < control). Detailed results of the main analysis are available in Table 3.

When excluding the participants with medical conditions or developmental disorders (n = 15), both intervention groups show significant differences in the change in HHb compared to the control (simple intervention: $B = 1.54 \times 10^{-07}$ mol/L, 95% CI [$0.16 \times 10^{-07}$, $2.91 \times 10^{-07}$], $p < 0.05$; cognitively engaging intervention: $B = 1.62 \times 10^{-07}$ mol/L, 95% CI

## All participants

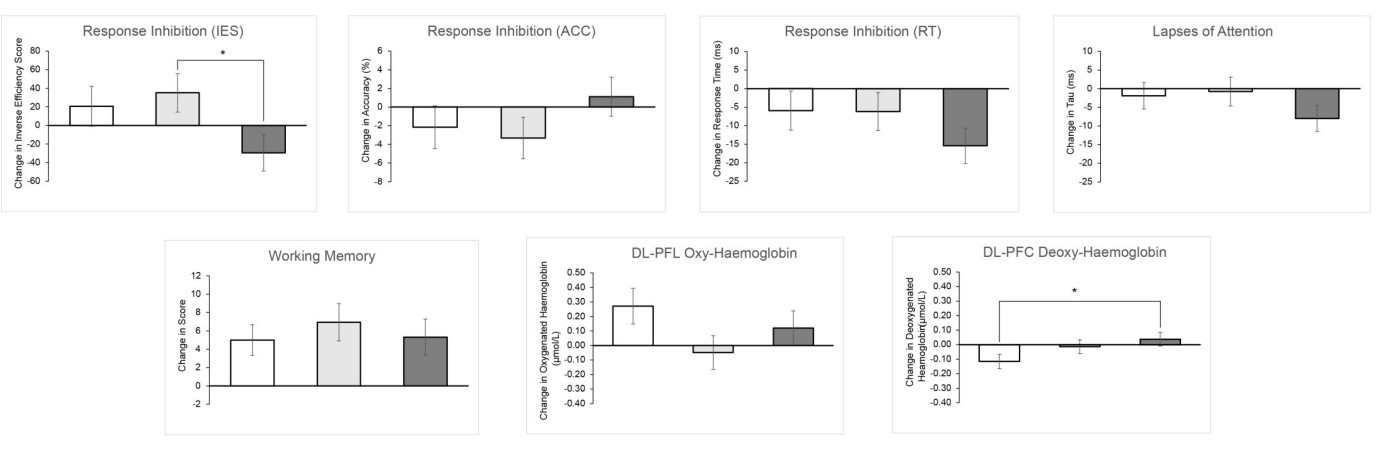

## Participants without medical conditions/disorders

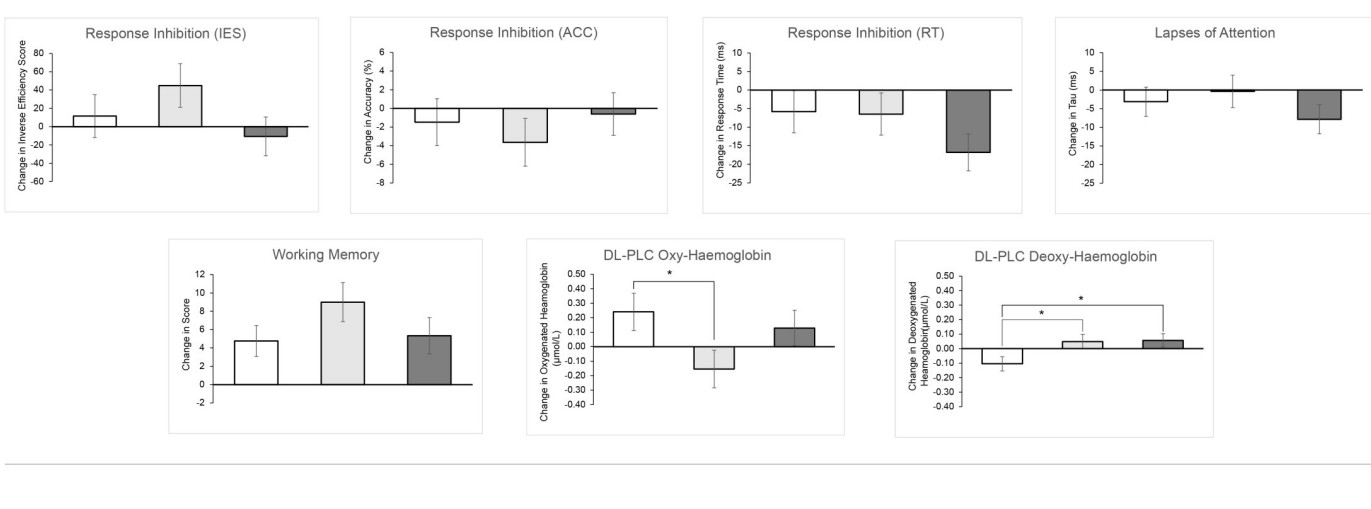

☐ Control ☐ Simple Active Breaks ■ Cognitively Engaging Active Breaks

**Fig 4. Effects of cognitively engaging and simple active breaks, and usual school practice, on children's cognitive functions and left dorsolateral prefrontal cortex haemodynamic response.** The graph shows the adjusted marginal change score means ± standard error for each study group—for the entire sample and for children who were not reported to have a medical or developmental condition. Inverse efficiency score (IES); response time (RT); accuracy (ACC); dorsolateral prefrontal cortex (DL-PFC).

[$0.28 \times 10^{-07}$, $2.95 \times 10^{-07}$], $p < 0.05$) and the simple active breaks also showed a significant reduction in the relative change of $O_2$Hb compared to the control ($B = -3.95 \times 10^{-07}$ mol/L, 95% CI [$-7.55 \times 10^{-07}$, $-0.36 \times 10^{-07}$], $p < 0.05$).

The observed effects of the study groups were negligible for working memory and lapses of attention (both $f^2 < 0.01$), small for inhibition inverse efficiency score, response time and accuracy ($f^2 = 0.04$, $f^2 = 0.02$ and $f^2 = 0.02$, respectively) and small to moderate for DL-PFC haemodynamic response (oxy: $f^2 = 0.05$; deoxy: $f^2 = 0.07$). A post-hoc analysis of the sample size revealed that the attained power was below the conventional 0.80 threshold for all main outcomes ($1 - \beta < 0.55$). The design effect did not influence most of the outcomes ($< 2$ conventionally indicated as the value below which the clustering in the data must not be accounted for), except for lapses of attention that showed a design effect of 2.73.

**Table 3. Intervention effects for cognitively engaging and simple active breaks compared to usual practice on children's cognitive functions and DL-PFC haemodynamic response [a].**

| Outcome variables | Baseline–End of trial (B [95% CI]) | | | Intervention Effect (B [95% CI]) | | | Residuals ICC [ICC 95% CI] |
|---|---|---|---|---|---|---|---|
| | Control | Simple active breaks | Cognitively engaging active breaks | Simple active breaks vs Control | Cognitively engaging active breaks vs Control | Cognitively engaging vs Simple active breaks | |
| **Cognitive functions** | | | | | | | |
| Response inhibition IES ($\Delta$ ms) (n = 141) | 20.61 [−21.53, 62.75] | 35.17 [−5.50, 75.83] | −29.42 [−67.78, 8.94] | 14.55 [−44.00, 73.11] | −50.03 [−107.20, 7.14] | −64.59 [*] [−120.58, −8.60] | $1.43 \times 10^{-19}$ [$1.43 \times 10^{-19}$, $1.43 \times 10^{-19}$] |
| *Response time ($\Delta$ ms) (n = 141)* | −5.92 [−16.24, 4.39] | −6.19 [−16.24, 3.85] | −15.40 [**] [−24.77, −6.03] | −0.27 [−14.77, 14.23] | −9.47 [−23.40, 4.45] | −9.20 [−23.02, 4.61] | $2.71 \times 10^{-22}$ [$2.71 \times 10^{-22}$, $2.71 \times 10^{-22}$] |
| *Accuracy($\Delta$ %) (n = 141)* | −2.15 [−6.64, 2.33] | −3.31 [−7.65, 1.03] | 1.12 [−2.98, 5.22] | −1.16 [−7.39, 5.08] | 3.27 [−2.81, 9.36] | 4.43 [−1.57, 10.43] | $3.12 \times 10^{-12}$ [$3.12 \times 10^{-12}$, $3.12 \times 10^{-12}$] |
| Lapses of attention ($\tau$) ($\Delta$ ms) (n = 141) | −1.91 [−8.82, 5.01] | −0.77 [−8.33, 6.80] | −7.97 [*] [−14.89, −1.06] | 1.14 [−9.12, 11.39] | −6.07 [−15.87, 3.73] | −7.21 [−17.45, 3.04] | 0.11 [0.02, 0.37] |
| Working memory ($\Delta$ score) (n = 77) | 5.00 [**] [1.72, 8.28] | 6.95 [**] [2.97, 10.93] | 5.32 [**] [1.47, 9.16] | 1.95 [−3.26, 7.15] | .32 [−4.73, 5.36] | −1.63 [−7.17, 3.91] | $6.40 \times 10^{-26}$ [$6.40 \times 10^{-26}$, $6.40 \times 10^{-26}$] |
| **DL-PFC haemodynamic response** | | | | | | | |
| O₂Hb ($\Delta$ mol/L) (n = 67) | $2.71 \times 10^{-07}$ [*] [$0.30 \times 10^{-07}$, $5.13 \times 10^{-07}$] | $−0.48 \times 10^{-07}$ [$−2.76 \times 10^{-07}$, $1.79 \times 10^{-07}$] | $1.20 \times 10^{-07}$ [$−1.10 \times 10^{-07}$, $3.50 \times 10^{-07}$] | $−3.20 \times 10^{-07}$ [$−6.54 \times 10^{-07}$, $0.15 \times 10^{-07}$] | $−1.51 \times 10^{-07}$ [$−4.92 \times 10^{-07}$, $1.89 \times 10^{-07}$] | $1.68 \times 10^{-07}$ [$−1.58 \times 10^{-07}$, $4.94 \times 10^{-07}$] | $1.23 \times 10^{-23}$ [$1.23 \times 10^{-23}$, $1.23 \times 10^{-23}$] |
| HHb ($\Delta$ mol/L) (n = 67) | $−1.15 \times 10^{-07}$ [*] [$−2.13 \times 10^{-07}$, $−0.17 \times 10^{-07}$] | $−0.14 \times 10^{-07}$ [$−1.06 \times 10^{-07}$, $0.78 \times 10^{-07}$] | $0.38 \times 10^{-07}$ [$−0.54 \times 10^{-07}$, $1.30 \times 10^{-07}$] | $1.01 \times 10^{-07}$ [$−0.35 \times 10^{-07}$, $2.37 \times 10^{-07}$] | $1.53 \times 10^{-07}$ [*] [$0.17 \times 10^{-07}$, $2.90 \times 10^{-07}$] | $−0.52 \times 10^{-07}$ [$−0.78 \times 10^{-07}$, $1.82 \times 10^{-07}$] | $9.39 \times 10^{-24}$ [$9.39 \times 10^{-24}$, $9.39 \times 10^{-24}$] |

DL-PFC, dorsolateral prefrontal cortex; CI, confidence interval; ICC, intraclass correlation coefficient; IES, inverse efficiency score; O₂HB, oxy-haemoglobin; HHB, deoxy-haemoglobin.

[a] Results of the main analyses using mixed models adjusted for sex, age, and baseline outcome value, and clustered at the classroom level; DL-PFC haemodynamic response models were also adjusted for the related cognitive performance. Being an inverse score, a lower IES indicates better cognitive performance.

[*] $p < 0.05$

[**] $p < 0.01$.

## Secondary analysis

**Sitting, standing, stepping patterns.** *Mid-trial.* The complete results of the analysis of sitting/standing/stepping patterns are available in S3 Table. Linear mixed models revealed no significant differences within or between groups in the change in class time spent sitting or stepping, at mid-trial. However, a significant reduction in the time spent in bouts of sitting greater than 5 or 20 minutes emerged for the control group (approximately −30 min and −33 min, respectively) and for children doing the simple active breaks (approximately −21 min and −17 min, respectively), but not for the cognitively engaging active breaks group. Compared to the control, the cognitively engaging active breaks appeared to have produced a significant negative effect (i.e., difference in the changes values) on sitting in bouts greater than 5 min (B = 35.69 min, 95% CI [12.20, 59.19], p = 0.003) and 20 min (B = 28.80 min, 95% CI [9.12, 48.49], p = 0.004).

There was a significantly higher number of in-class sit-to-stand transitions at mid-trial compared to baseline for the control (approximately +9) and the simple intervention (approximately +7), but not the cognitively engaging group. This resulted in a negative effect of the cognitively engaging intervention on sit-to-stand transitions compared to the control (B = −10.28, 95% CI

[–14.71, –5.85], $p < 0.001$). Moreover, the cognitively engaging intervention group showed a significant reduction in the class time spent standing at mid-trial compared to baseline ($B = -6.25$ min, 95% CI [–12.18, –0.32], $p = 0.039$), although the overall effect of this reduction was not significant compared to the control group.

Similar results were found in the change values between baseline and mid-trial during school time (i.e., including recess and lunch).

*End of trial.* By the end of trial, the control group showed a significant increase in class time sitting compared to the baseline value (approximately +11 min), while this appeared relatively stable for both the intervention groups (S3 Table). Thus, the difference in the change values in class time sitting between the simple and the control group ($B = -13.53$ min, 95% CI [–26.59, –0.48], $p = 0.042$) and between the cognitively engaging intervention and the control group ($B = -13.47$ min, 95% CI [–26.22, –0.71], $p = 0.038$) appeared both statistically significant, showing a positive effect of interventions. Similarly, the control group spent significantly more time in sitting bouts greater than 20 min (approximately +15 min) compared to baseline, whereas no significant differences emerged for the intervention groups. Thus, the simple intervention ($B = -23.94$ min, 95% CI [–41.79, –6.09], $p = 0.009$) and the cognitively engaging intervention ($B = -18.93$ min, 95% CI [–36.22, –1.63], $p = 0.032$) had a positive effect on sitting for more than 20 min, compared to the control.

All groups showed significant increases in the number of class time sit-to-stand transitions performed at end of trial compared to baseline, therefore no significant differences emerged from a between-group comparison. The cognitively engaging intervention spent significantly more class time stepping (approximately +2 min) and accumulated a significant higher number of steps (approximately +173 steps), at the end of trial compared to baseline. This resulted in positive effects of the cognitively engaging intervention on stepping time and total step count ($B = 3.78$ min, 95% CI [0.64, 6.92], $p = 0.018$ and $B = 264.86$ min, 95% CI [67.16, 462.56], $p = 0.009$, respectively) compared to the control group.

**Mediation analysis.**   Since we did not find significant indirect effects for lapses of attention, working memory or DL-PFC haemodynamic response we have only reported the effects of the study conditions on response inhibition IES (Table 4). For both intervention conditions compared to the control, we found significant natural indirect effects (NIE) on the baseline–end of trial change in response inhibition IES mediated by sitting/standing time, with the cognitively engaging condition showing greater effects (NIE = –31.26, 95% CI [–93.52, –1.11], $p = 0.036$, and NIE = –26.28, 95% CI [–74.54, –2.60], $p = 0.048$, respectively; Fig 5) than simple active breaks condition (NIE = –15.63, 95% CI [–46.76, –0.56], $p = 0.036$, and NIE = –13.14, 95% CI [–37.27, –1.30], $p = 0.048$, respectively).

While the effects on response inhibition were apparently only explained by the indirect pathways (i.e., no significant direct effects were observed), the cognitively engaging intervention showed significant exposure-mediator interactions with the change in time spent sitting or standing, meaning that the effects of cognitively engaging active breaks on response inhibition varied according to the varying levels of change in sitting/standing time (Fig 6).

**On-task behaviour.**   Table 5 presents the percentages of observation rated as on-task in each observation period divided by group. The results of the multilevel generalised linear mixed models (Table 6) show that, relative to the control group, the simple active breaks group had an 87% reduction in the odds of being observed on-task in the second 30-min observation period compared to the first (odds ratio [OR] = 0.13, 95% CI [0.05, 0.31], $p = 0.001$), at baseline. No statistical differences were noted between the control and the cognitively engaging group at this time point. At mid-trial, the simple active breaks group had 60% lower odds than the control of being observed on-task during the second in-class observation period compared to the first (OR = 0.40, 95% CI [0.18, 0.93], $p < 0.05$); the differences between the control and

**Table 4. Direct, indirect, and total effects of cognitively engaging/simple active breaks on response inhibition.**

| Mediators | P-value (exposure-mediator interaction) | Response Inhibition IES, coefficient [bootstrapped 95% CI] | | | | | |
|---|---|---|---|---|---|---|---|
| | | NDE | NIE | TE | CDE (mediator – 1SD) | CDE (mediator M) | CDE (mediator +1SD) |
| **Cognitively engaging vs Control** | | | | | | | |
| *Without interaction term* | | | | | – | – | – |
| Sitting time change | – | −5.29 [−74.67, 63.68] | −31.26 * [−93.52, −1.11] | −36.55 [−113.49, 33.16] | – | – | – |
| Standing time change | – | −10.46 [−79.37, 55.53] | −26.28 * [−74.54, −2.60] | −36.74 [−114.49, 32.32] | – | – | – |
| Stepping time change | – | −24.74 [−92.34, 41.74] | −11.34 [−63.11, 14.33] | −36.08 [−113.49, 33.16] | – | – | – |
| *With interaction term* | | | | | | | |
| Sitting time change | 0.002 * | 27.57 [−51.91, 114.33] | −71.35 * [−177.91, −20.79] | −43.78 [−122.29, 22.20] | −158.19 * [−282.33, −58.87] | −27.21 [−100.58, 35.58] | 103.78 [−1.78, 212.24] |
| Standing time change | 0.003 * | 5.98 [−67.07, 84.47] | −60.02 * [−147.40, −19.21] | −54.04 [−132.99, 11.34] | 90.15 [−7.21, 185.71] | −41.01 [−114.47, 19.63] | −172.16 * [−301.00, −71.56] |
| Stepping time change | 0.078 [a] | 15.34 [−70.55, 116.89] | −36.77 [−131.49, 15.03] | −21.43 [−89.84, 42.83] | 64.56 [−74.89, 237.42] | −13.24 [−85.49, 55.91] | −91.65 [−236.07, 27.04] |
| **Simple intervention vs Control** | | | | | | | |
| *Without interaction term* | | | | | | | |
| Sitting time change | – | −2.64 [−37.33, 31.84] | −15.63 * [−46.76, −0.56] | −18.28 [−56.74, 16.58] | – | – | – |
| Standing time change | – | −5.23 [−39.68, 27.76] | −13.14 * [−37.27, −1.30] | −18.37 [−57.24, 16.16] | – | – | – |
| Stepping time change | – | −12.37 [−46.17, 20.87] | −5.67 [−31.56, 7.16] | −18.04 [−56.74, 16.58] | – | – | – |

IES, inverse efficiency score; CI, confidence interval; NDE, natural direct effect; NIE, natural indirect effect; TE, total effect; CDE, controlled direct effect; SD, Standard deviation; M, mean. A negative change in IES is indicative of improved performance at the cognitive task. All models were adjusted for adjusted for child sex and age. For the simple active break intervention, the models with exposure-mediator interactions were not reported as all the interactions were not significant.

[a] $p = 0.05–0.10$

* $p < 0.05$.

the cognitively engaging group during the two observation periods appeared not statistically significant at this time point. Although teachers were supposed to break up the lesson with an active break at this time point, the observers noted that the active breaks were not conducted between the two consecutive in-class observations. At the end of trial, the simple active breaks and the cognitively engaging group showed higher odds than the control (127% and 178%, respectively) of being observed on-task during the second in-class observation period, compared to the first (OR = 2.27, 95% CI [1.02, 5.08], $p < 0.05$ and OR = 2.78, 95% CI [1.16, 6.66], $p < 0.05$, respectively). This time, the active breaks were conducted between the two in-class observations as per intervention intention.

**Children's enjoyment, and physical/mental effort.** At post-intervention, children in both intervention groups reported high levels of enjoyment related to the active breaks (simple intervention: mean [M] = 4.19, standard deviation [SD] = 0.5; cognitively engaging intervention: M = 4.18, SD = 0.58; max possible score = 5), which did not appear different by group (t(93) = 0.1, p = 0.92). Similarly, the levels of physical exertion reported by children in the simple active breaks (M = 2.54, SD = 2.61) were comparable to the ones reported by the cognitively engaging group (M = 2.80, SD = 2.41), t(93) = −0.50, p = 0.62. Children in the simple

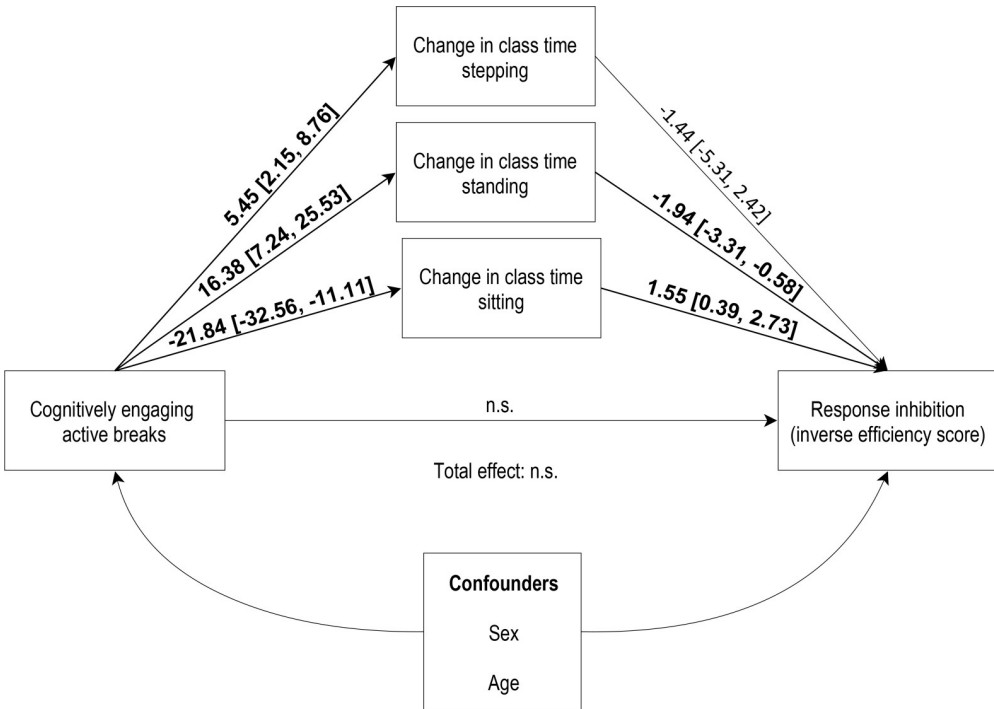

**Fig 5. Mediated effects of cognitively engaging active breaks on response inhibition inverse efficiency score via changes in sitting, standing, and stepping time.** Natural indirect effects (NIE) of cognitively engaging active breaks on response inhibition. Results are presented as *B* (95% CI). Significant effects in bold. Natural direct effects (NDE) and total effects were non-significant (n.s.).

intervention group reported similar levels of cognitive engagement (M = 2.27, SD = 2.88) to what was expressed by children in the cognitively engaging group (M = 2.47, SD = 2.00), t(93) = −0.39, p = 0.69.

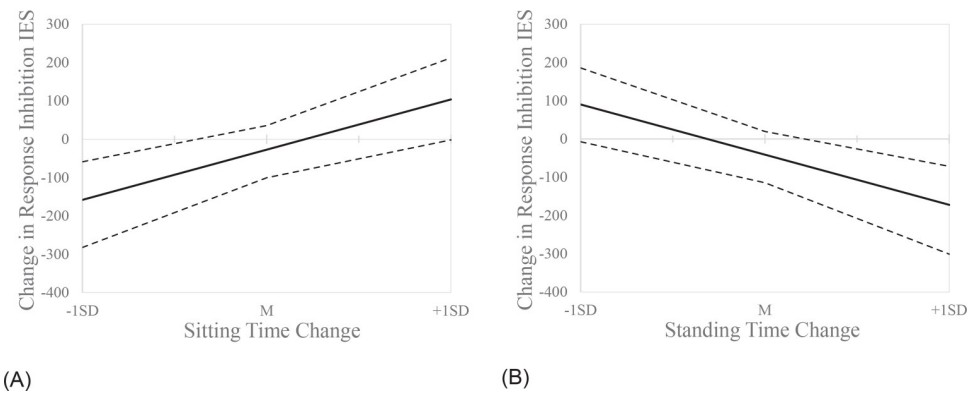

(A)                                                         (B)

**Fig 6.** Interactive effects of the cognitively engaging active breaks at different levels of sitting (A) and standing time change (B). The solid lines represent the estimated values. The dotted lines show the upper and lower bootstrapped 95% confidence intervals. A response inhibition inverse efficiency score (IES) change is indicative of improved performance at end of trial compared to baseline. The plots show that when there was a reduction in the time spent sitting or an increase in the time spent stepping (by the end of trial compared to baseline) children doing the cognitively engaging active breaks had a significant positive effect on response inhibition (i.e., negative change in IES score). Conversely, the opposite effect was observed in case of a change in sitting time greater than +1 standard deviation (SD) (i.e., baseline–end Δ > 27.2 min) or a change in standing time lower than −1SD (i.e., baseline–end Δ < −23.2 min).

**Table 5. Percentage of observations rated as on-task across the observation periods by group.**

| Group | Baseline | | Mid-trial | | End of trial | |
|---|---|---|---|---|---|---|
| | 30-min pre | 30-min post | 30-min pre | 30-min post | 30-min pre | 30-min post |
| Control (%) | 64.2 | 74.6 | 70.0 | 78.2 | 75.0 | 51.1 |
| Simple (%) | 87.1 | 59.1 | 77.5 | 68.1 | 63.3 | 56.7 |
| Cognitively engaging (%) | 68.3 | 68.3 | 76.7 | 75.8 | 79.5 | 78.3 |

At each study time point (baseline, mid-trial, end of trial), two consecutive 30-min in-class observations (pre- post- observation periods) were conducted. During each observation period, six children from one classroom per study group were systematically observed during their lesson for 20 intervals per child (each lasting 10 seconds). The observer marked each interval as on-task or off-task depending on children's behaviour during the interval.

**Intervention fidelity.** Activity logs were collected from teachers and suggested that most activities were implemented as prescribed. However, the analysis of children's sedentary patterns suggested that significant changes were only observed by the end of trial and not at mid-trial. It is possible that this finding is reflective of the time teachers need to familiarise with new activities before implementing them regularly.

## Discussion

The primary aim of the study was to investigate the effects of simple and cognitively engaging active breaks on children's cognitive functions and DL-PFC haemodynamic response. To our knowledge, this is the first study to have ever tested any type of classroom-based physical activity on children's response inhibition, lapses of attention and DL-PFC haemodynamic response.

The results from mixed models showed that simple or cognitively engaging active breaks conducted for six weeks did not significantly affect children's cognitive functions compared to the usual practice control group, although within group improvements in response inhibition response time and lapses of attention were noted for children in the cognitively engaging intervention by the end of trial. Our fNIRS data showed that children in the control condition showed significant reductions in HHb and $O_2$Hb over time, whereas no significant changes were observed for the intervention groups. A significant positive effect of the cognitively challenging active breaks was observed on HHb compared to the control group. As hypothesised, we expected children in these groups to show a greater reduction in $O_2$Hb or increase in HHb change levels compared to the control group, under the same level of cognitive performance. This is suggestive of more efficient neural activity in the DL-PFC due to the lower level of metabolic resources (haemodynamic response) necessary to achieve the same level of cognitive

**Table 6. Results from generalized linear mixed models clustered at the individual level, conducted to investigate the group differences in the odds ratios of being observed on-task during two consecutive in-class observations at each time point.**

| Predictor | Baseline | | | Mid-trial | | | End of trial | | |
|---|---|---|---|---|---|---|---|---|---|
| | OR | [95% CI] | P-value | OR | [95% CI] | P-value | OR | [95% CI] | P-value |
| Simple active breaks [a] | 0.13 | [0.05, 0.31] | 0.001** | 0.40 | [0.18, 0.93] | 0.033* | 2.27 | [1.02, 5.08] | 0.045* |
| Cognitively engaging breaks [a] | 0.61 | [0.28, 1.34] | 0.218 | 0.62 | [0.27, 1.44] | 0.268 | 2.78 | [1.16, 6.66] | 0.022* |

OR, odds ratios; CI, confidence interval.

[a] Coefficients represent the difference in odds ratio of being observed on-task for each intervention group against the control between two consecutive 30-min observation periods at each study time point.

* $p < 0.05$

** $p < 0.01$

performance [66]. There was a marginally significant negative change in $O_2Hb$ for the simple active break intervention only, but the current finding of a greater positive change in HHb for the cognitively engaging intervention carries the same meaning in terms of efficiency. Although the right DL-PFC has been previously indicated as the neural substrate of inhibition [83], a study conducted in adults revealed that better efficiency at the Go/No-Go task was correlated with left-hemispheric dominance of the DL-PFC [84], demonstrating a neural basis of this connection. Some researchers have hypothesised that the left-hemisphere may have a supporting role during complex inhibition tasks that fully engage the right-hemisphere [85], or simply help right DL-PFC to direct the attention on the tasks that requires inhibition [86].

It is, however, unlikely that a clear lateralisation of the DL-PFC exists in young children, considering that previous research suggested that the development of the DL-PFC occurs throughout childhood though to early adulthood [87,88]. Thus, it is possible that the left and right DL-PFC act in unity as the neural substrate of response inhibition in children. Patterns of $O_2Hb$ and HHb often appear in alternation, meaning that an increase in $O_2Hb$ is accompanied by a reduction in HHb and vice versa [66]. Also, the algorithms used to calculate $O_2Hb$ and HHb are based on different wavelengths (usually 760 and 850 nm, respectively) [66]. The resulting signals in response to neural activity typically show more sensitive changes in $O_2Hb$ compared to HHb [66]. However, the HHb signal was found to have better spatial specificity compared to $O_2Hb$ [89], which might partially explain our significant findings in relation to the HHb but not $O_2Hb$. Despite the absence of statistical significance, the results of the mixed models show that both active conditions, compared to the control, had negative average changes in $O_2Hb$ (although not statistically significant) and positive average changes in HHb (only significant for the cognitively engaging group) while controlling for the cognitive performance at the concurrent cognitive test, which aligns with our hypothesis of improved DL-PFC efficiency for children in the cognitively engaging intervention. Thus, our findings on DL-PFC haemodynamic response partially confirms the "cognitive stimulation hypothesis", according to which the effects of cognitively engaging physical activity might be greater than simple activities because they activate the same brain area involved in higher-order thinking [9,90,91].

While some studies have shown that active breaks can lead to improvements in children's attention [43,44,92,93], executive functions [43], and academic achievement [e.g., 94], our findings only partially supported evidence of cognitive benefits from active breaks. Only three studies have examined the effects of cognitively engaging active breaks to investigate the effects on primary school children's cognition/learning [45,56,57]. Schmidt et al. [45] tested the acute effects of four 10-min types of classroom-based active breaks (i.e., high-high, high-low, low-high, and low-low orthogonal combinations of physical and cognitive engagement) on attention and processing speed, finding that cognitive engagement was the only factor positively related to the cognitive outcomes. A more recent acute study [57] adopted a similar approach to explore the acute effects of 6-min active breaks with different levels of physical and cognitive engagement on children's executive functions, and found that cognitive engagement could deteriorate children's cognitive performance in shifting (i.e., one of the executive functions), with no effects found on other executive functions (i.e., inhibition and working memory). Interestingly, another recent study from the same authors [56] tested the chronic effect of 10-min active breaks over 20 weeks finding that cognitively engaging active breaks may have a positive acute effect on children's shifting ability; no effects were found on inhibition or working memory. Whilst we also did not find effects on working memory, our study provided evidence that active breaks can enhance response inhibition. It is important to note that the measure of inhibition used in Egger and colleagues' studies [56,57] was a task designed to assess the participant's ability to stay focused on a visual cue when some interfering cues are presented (inhibition of interference). Instead, we assessed the ability to refrain responses

when required (response inhibition), which represents a different facet of the construct of inhibitory function [3].

As highlighted above, active breaks, including cognitively engaging classroom-based strategies, can facilitate certain aspects of children's cognitive functioning and facilitate children's learning. However, collectively, the effects of this approach on children's attention and executive functions still unclear. Perhaps, this might be explained by the recent review by Mavilidi et al. [95], who proposed a combination of the effective findings from exercise and cognition research with the embodied cognition approaches. Exercise and cognition research is focused on the acute/chronic effects of exercise on cognitive functions, using high intensity physical activities that are normally of low relevance in the educational context (e.g., running). On the other hand, embodied cognition research is generally characterised by the use of low intensity gestures, which are highly relevant to the learning outcome of interest (e.g., the enactment of a concept using movement). Mavilidi et al. [95] proposed a joint classification of exercise and cognition studies and embodied cognition studies by referring to two dimensions: the extent to which physical activities are integrated with and relevant to learning subjects. Thus, it is possible that the different levels of integration and relevance of the cognitively challenging activities presented above provide some justification for the different findings. A blend of the best qualities of the two approaches could combine the physiological and cognitive stimulation and may result in activities that support children's cognitive development and learning outcomes more than traditional approaches. This, however, would need to be tested empirically in a laboratory as well as in an ecological learning context to be confirmed.

No studies have previously explored whether the effects of active breaks on cognition could be mediated by a change in the time spent sitting, standing, or stepping. Although the results from our main analysis provided limited evidence of the effects of the interventions on children's cognition, the results from the mediation models suggest that the intervention had positive effects on response inhibition via a reduction in sitting time and/or an increase in standing time. In our study, when cognitively engaging active breaks effectively reduced class time sitting (negative change) and/or increased standing (positive change), significant direct and indirect positive effects on response inhibition were observed (i.e., negative change in the IES, meaning that the performance improved over time). Conversely, an increase in time spent sitting by more than one standard deviation from the mean (SD = 27.2 min) and/or a reduction in the time spent standing by less than one standard deviation from the mean (SD = –23.2 min), corresponded to worsened response inhibition performance.

Although the sitting/standing/stepping patterns did not change at mid-trial, implementation fidelity was observed by the end of the trial with significant improvements in these patterns. Previous research [32,96,97] suggests that teachers perceive limited availability of time, in an already busy curriculum, as a major barrier to the implementation of active breaks, whereas increased confidence in implementing the program, gained through training or experience, is seen as a facilitator. Therefore, it is possible that teachers in our intervention groups had to familiarise themselves with the active breaks and make sure to have these activities in their schedules before being able deliver them regularly twice a day, as planned.

Previous research that examined the effects of active breaks on on-task behaviour [49–51] generally found positive acute effects on children. By the end of the current trial, the active conditions in our study showed significant higher odds of being observed on-task compared to the control group. Children in the intervention groups showed similar and high levels of enjoyment, which confirms that children generally enjoy classroom-based physical activity [32,42]. In line with previous research [38], our study found that children doing the active breaks reported high levels of enjoyment in relation to these activities. As hypothesised, no differences in children's enjoyment were noted between simple and cognitively engaging active

breaks. The notion that a comfortable learning environment that stimulates enjoyment and discovery can facilitate and affect cognitive functions and learning is not new [42]. Thus, our finding regarding enjoyment may favour the use of active breaks in the classroom.

The manipulation check of the physical and cognitive effort required to participate in the active breaks confirmed our hypothesis that the two interventions were comparable in terms of physical intensity. Despite the fact the two interventions were designed to have different levels of cognitive engagement, children's perception of the cognitive effort required to conduct the active breaks was comparable between the two groups. We might have failed to appropriately manipulate the cognitive engagement component of the activities or, perhaps more likely, the measure used to test cognitive engagement may not effectively measure children's perceptions on this aspect, as the validity of this measure has not been tested.

One of the main strengths of this study is the employment of objective measures of sitting time using inclinometers that are considered more suitable than other activity monitors (e.g., accelerometers) in capturing sedentary behaviour patterns. Other strengths include the use of an fNIRS device to objectively measure DL-PFC haemodynamic response and the large sample size for a brain imaging study. Previous studies using this technique have rarely targeted children and have had sample sizes between 20 to 50 participants [98]. Another value add of this study is the combination of behavioural and brain activity data, which permits to examine cognition in its entirety and might allow to explain the neural mechanisms underlying certain observed behavioural patterns.

It should be noted that this study comes with some limitations. Although we randomly allocated the intervention conditions, we could not randomly assign classrooms to the control group. Perhaps, one of the issues with recruiting schools relates to the number of assessments that are needed to be completed. Given that the participants for this study were contacted using convenience sampling, the findings of this study might not be generalizable to all 6–8-year-olds. It is possible that the study was not sufficiently powered to detect significant differences in all outcomes of interest, particularly for the cognitive tests that involved a reduced sample (i.e., working memory and fNIRS). The post-hoc power calculation based on the observed effects revealed that the sample size in our study was potentially small to allow us detecting significant differences. Despite the limited power, our study detected a significant effect of the cognitively engaging active breaks in the proportion of deoxygenated haemoglobin in the left dorsolateral prefrontal cortex. Thus, it provided useful information on the higher sensitivity of neural as compared to behavioural cognitive measures in this type of ecological school-based intervention. Also, the provided effect sizes are useful for estimating sample size to conduct future well-powered studies. It is worth noting that the observed effect sizes in the main outcome were smaller than the effects generally observed in previous studies [e.g., 56], presumably depending on implementation factors. It is possible that the breaks were too short or not sufficiently frequent/intense to generate the expected cognitive response in children. Indeed, the results from our mediation analysis showed that response inhibition was affected by the active breaks depending on the extent of change in sitting and standing time elicited by the breaks. An additional limitation is that we did not assess children's sleep, energy intake/ diet, and weight status, which could have partially explained variability in children's executive functions [99,100]. It is important to note, however, that compared to the fully controlled environment which applies to a laboratory setting, conducting real-world research has its own complexity because of the many environmental factors that are difficult to control [101]. The results from our working memory assessment showed that all children improved their performance getting close to the highest possible score, suggesting that the measures used might not be sufficiently discriminatory. Moreover, children's sitting data were collected on days not including physical education or school sport, to avoid the confounding effect of those activity, but we did not use the same approach when measuring cognitive functions. To avoid excessive

disruption to the school, sitting and cognition were measured on different days within the same week, so it is possible that children had physical education or school sport when we measured cognitive functions. This might have resulted in a small positive acute effect on children's cognition [102]. Another limitation is the employment of a single channel fNIRS device, which has limited spatial resolution compared to a whole-head system.

## Conclusion

Our study adds to the literature with suggestive evidence supporting the notion that cognitively engaging active breaks may help to improve children's neural activity efficiency in the DL-PFC (neural substrate of executive functions) and response inhibition abilities. However, neither intervention showed significant improvements in working memory or lapses of attention, compared to control. We also showed that the intervention had positive effects on response inhibition via a reduction in sitting time and/or an increase in standing time.

Given that the traditional teaching approach is still largely sedentary [103], identifying new ways to reduce sitting and increase physical activity in this population appears important for children's physical health and might also benefit their cognitive functions. Expecting children to meet the physical activity and sedentary behaviour guidelines, might be unrealistic if they are not also provided with opportunities to be active throughout the day. Our study and other research suggest that active breaks are feasible to implement in primary schools [32].

Future studies could consider active breaks of longer duration or frequency, and more aligned with the educational context, as they might show greater effects on children's cognitive functions. Also, it would be important to measure sleep and weight status as potential confounders. Randomised controlled trials with large samples using objective measures similar to the ones we used are needed. Finally, the use of a more comprehensive cognitive assessment batteries and multi-channel brain imaging devices are recommended.

## Supporting information

**S1 Checklist. TREND checklist.**
(PDF)

**S1 Table. Simple and cognitively engaging intervention activities.**
(PDF)

**S2 Table. Haemodynamic response data processing pipeline and MATLAB processing script.**
(PDF)

**S3 Table. Results from linear mixed models, used to understand differences by study group on the change in sitting, standing, and stepping patterns.** All models were adjusted for the baseline value related to each outcome variable, child sex and age, and controlled for the random effects of classroom as a clustering variable.
(PDF)

**S1 Data. Trial study protocol.**
(PDF)

## Acknowledgments

Ms Elizabeth Vella and Ms Sacha Bosman assisted with the data collection. Prof Liliana Orellana advised on data analysis. Dr Winda Liviya Ng assisted with the mediation analysis. We

sincerely thank consenting schools, teachers, children, and parents for their participation. This manuscript has been completed as part of PhD requirements.

## Author Contributions

**Conceptualization:** Emiliano Mazzoli, Jo Salmon, Wei-Peng Teo, Lisa Michele Barnett.

**Data curation:** Emiliano Mazzoli.

**Formal analysis:** Emiliano Mazzoli, Wei-Peng Teo.

**Funding acquisition:** Emiliano Mazzoli, Lisa Michele Barnett.

**Investigation:** Emiliano Mazzoli, Jo Salmon, Lisa Michele Barnett.

**Methodology:** Emiliano Mazzoli, Jo Salmon, Wei-Peng Teo, Caterina Pesce, Jason He, Lisa Michele Barnett.

**Project administration:** Emiliano Mazzoli, Lisa Michele Barnett.

**Resources:** Emiliano Mazzoli, Wei-Peng Teo, Caterina Pesce, Lisa Michele Barnett.

**Software:** Emiliano Mazzoli, Wei-Peng Teo, Jason He.

**Supervision:** Jo Salmon, Wei-Peng Teo, Caterina Pesce, Lisa Michele Barnett.

**Validation:** Emiliano Mazzoli, Jo Salmon, Wei-Peng Teo, Caterina Pesce, Jason He, Tal Dotan Ben-Soussan, Lisa Michele Barnett.

**Visualization:** Emiliano Mazzoli.

**Writing – original draft:** Emiliano Mazzoli.

**Writing – review & editing:** Emiliano Mazzoli, Jo Salmon, Wei-Peng Teo, Caterina Pesce, Jason He, Tal Dotan Ben-Soussan, Lisa Michele Barnett.

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
