## [Decision Letter · Decision Letter 0]

10 Feb 2021

PONE-D-20-22144

Breaking up classroom sitting time with cognitively engaging physical activity: behavioural and brain responses

PLOS ONE

Dear Dr. Mazzoli,

Thank you for submitting your manuscript to PLOS ONE. After careful consideration, we feel that it has merit but does not fully meet PLOS ONE’s publication criteria as it currently stands. Therefore, we invite you to submit a revised version of the manuscript that addresses the points raised during the review process.

Both reviewers expressed several concerns regarding your methodology, statistical analysis, results and their interpretation, and conclusions.

We look forward to receiving your revised manuscript.

Kind regards,

Maciej S. Buchowski

Academic Editor

PLOS ONE

2. Our internal editors have looked over your manuscript and determined that it is within the scope of our Cognitive Developmental Psychology Call for Papers. The Collection will encompass a diverse range of research articles in developmental psychology, including early cognitive development, language development, atypical development, cognitive processing across the lifespan, among others, with an emphasis on transparent and reproducible reporting practices.  Additional information can be found on our announcement page: https://collections.plos.org/s/cognitive-psychology.  If you would like your manuscript to be considered for this collection, please let us know in your cover letter and we will ensure that your paper is treated as if you were responding to this call. Please note that being considered for the Collection does not require an additional peer review beyond the journal’s standard process and will not delay the publication of your manuscript if it is accepted by PLOS ONE. If you would prefer to remove your manuscript from collection consideration, please specify this in the cover letter.  

3. Thank you for submitting your clinical trial to PLOS ONE and for providing the name of the registry and the registration number. The information in the registry entry suggests that your trial was registered after patient recruitment began. PLOS ONE strongly encourages authors to register all trials before recruiting the first participant in a study.

1) your reasons for your delay in registering this study (after enrolment of participants started);

2) confirmation that all related trials are registered by stating: “The authors confirm that all ongoing and related trials for this drug/intervention are registered”.

Please also ensure you report the date at which the ethics committee approved the study as well as the complete date range for patient recruitment and follow-up in the Methods section of your manuscript.

"I have read the journal's policy and the authors of this manuscript have the following competing interests: JS declares that she has a potential conflict of interest as her husband established a business to manufacture height-adjustable desks for schools in 2017. However, height-adjustable desks were not used in this study and she was not involved in the data analysis. The other authors declared no competing interests. "

Reviewers' comments:

Reviewer's Responses to Questions

**Comments to the Author**

1. Is the manuscript technically sound, and do the data support the conclusions?

Reviewer #1: No

Reviewer #2: Partly

2. Has the statistical analysis been performed appropriately and rigorously? 

Reviewer #1: No

Reviewer #2: Yes

3. Have the authors made all data underlying the findings in their manuscript fully available?

Reviewer #1: No

Reviewer #2: Yes

4. Is the manuscript presented in an intelligible fashion and written in standard English?

Reviewer #1: Yes

Reviewer #2: Yes

5. Review Comments to the Author

Reviewer #1: A cluster randomized controlled trial was conducted to investigate the effects of simple intervention, cognitively engaging active break intervention or control on children’s cognitive function and brain activity. Cognitively engaging active breaks produced a significant negative effect on sitting bouts greater than 5 minutes over mid and final time points and sit-to-stand transitions at mid-trial. The simple intervention and the cognitively engaging intervention showed a positive effect on sitting for 20 minutes or more, compared to the controls.

Major revisions:

1- Table 5: Include time point in the model. If the interaction effect is significant, provide an interpretation of the results, but do not test main effects because the tests for main effects are uninteresting in light of significant interactions. If interaction effects are non-significant, drop the interaction effects from the model and test the main effects. Determining which results to present when testing interactions is often a multi-step process.

2- State and justify the study’s target sample size with a pre-study statistical power calculation. The power calculation should include: (1) the estimated outcomes in each group; (2) the α (type I) error level; (3) the statistical power (or the β (type II) error level); (4) the target sample size and (5) for continuous outcomes, the standard deviation of the measurements.

Minor revisions:-

1- Abstract: Indicate that the schools were randomly assigned to intervention.

2- Line 349: State the underlying covariance structure used in the linear mixed models and the criteria for selecting it.

3- Line 527 and beyond: Clarify what summary statistic is represented by the letter M.

4- Table 1: In the statistical analysis section, include the statistical methods used to compare groups in table 1.

Reviewer #2: Breaking up classroom sitting time with cognitively engaging physical activity: Behavioral and brain responses. In this research, the authors examined a maximum of 141 children between the ages of 6 – 8 years old that participated in a simple or a cognitively engaging active break intervention for 6 weeks or were part of a control group. Baseline and post-intervention measurements were taken of response inhibition, attentional lapses, working memory, and relative changes in brain hemodynamic response to determine the effect of the interventions. Differences in HHB in the cognitively engaged group suggested increased brain efficiency in that group, although no groups differences existed in working memory or lapses of attention. Response inhibition improved in the cognitively engaging active breaks group.

With this study, the authors ask an important question: What are the impacts of different types of activity breaks on measures important to children’s success in the classroom? While it is tempting to believe that cognitively engaging active breaks would be of greater benefit to children, the topic itself has received very little empirical attention. The current study goes further and attempts to look at underlying mechanisms (or at least differences in brain oxygenation) that may change as a result of the different types of interventions. Overall, the study is ambitious and the design is appropriate. My biggest concerns are related to the fNIR data and the conclusions drawn from hemodynamic measures, particularly considering that the key conclusions the authors draw come from these data. The use of fNIR data to quantify group differences is innovative and interesting. At the same time, data collection, analysis, and interpretation can be fraught with error and therefore, requires documentation and detail to ensure the process was performed in a way that ensured the best data fidelity. Overall, the methodology is lacking in specific details related to the use of the Artinis PortaLite system. Based on the description of the fNIR system, it provided hemodynamic information from a single channel fitted to the left prefrontal cortex for each child. Who placed the sensor and how was the location above the left prefrontal cortex verified? With EEG, a 10 – 20 placement system is frequently used to standardize the location of electrode placement; how did the authors ensure standardized placement, particularly considering the smaller size of children’s foreheads? Was the skin prepared prior to placement? How was the signal optimized (e.g., noise reduction) for each participant, given that individual differences exist in skin transparency based on skin color? There is almost no information about signal processing. How were motion artifacts detected and removed? The authors should provide this and other information related to the fNIR system, particularly because many readers interested in this research may not be familiar with the technology.

Another issue comes from the interpretation of the fNIR data. The authors collected data from the left PFC, which is a relatively small brain region involved in multiple cognitive activities. At the same time, any individual cognitive activity consists of a functional network of interacting brain structures, so there is not necessarily a clear correlation between a ‘greater positive change in the proportion of HHB’ that the cognitively active children demonstrated and an increase in cognitive efficiency. Also, in many studies, O2Hb is the more sensitive biomarker; the authors may want to discuss why that measure does not replicate the HHB results (and could potentially describe the differences in the two measures). In that regard, the fNIR results should be more speculative, and perhaps be a secondary focus of the study rather than a primary focus.

Below are specific questions or comments within each section of the manuscript.

ABSTRACT

Line 5 and 8. The authors refer to ‘brain activity’ in general. They should be more specific here to the prefrontal cortex, since that is the only measure of brain activity (more specifically, brain oxygenation), that they are observing.

Line 11. Age range of participants in abstract (6 – 8 years) is different than in text (6 – 9 years, on page 8)

Line 11. The reported sample size is misleading, especially considering that for working memory and fNIR assessments, the authors used about half that number. Provide a little more detail on participant number per analysis somewhere in the methods.

Line 17. It is probably more appropriate to use the term “haemodynamic response” rather than “brain activity”

Line 29. I recommend using the word “suggestive” rather than “indicative” to be more speculative.

Line 37. This conclusion seems appropriately speculative.

KEY POINTS

Line 45, # 1. I would argue that the authors have not proved this point and should not list it as their first key point.

INTRODUCTION

Overall, the introduction is well written, but needs more information on Haemodynamics of the brain and what that means.

Line 123. Somewhere in this section, the authors should expand their discussion of neural measures and haemodynamic response. For example, there is no mention of O2Hb or HHB (what they measure, how they change with neural activity, what they might mean). The authors could also introduce other research that uses these measures as an indication of cognitive efficiency.

Line 131. State how long the intervention took place.

Line 132. What does “normal practice” mean?

Line 142. “higher” efficiency, not “better” efficiency.

METHOD

Line 169. Ages are different than reported in Abstract.

Line 222. Did the authors check for fidelity of teachers presenting intervention?

Line 264, Is the Working Memory validated for 6 years old?

Line 270 – 284. Expand this section (see above). It may be useful to include a figure depicting a participant with the Artinis system on.

Line 354. Were all the variables normally distributed? If not, were any adjustments made?

RESULTS

Overall, the results were presented in a way that is easy to follow and understand.

Line 449 – 456. The authors might consider using graphs to present the O2Hb and HHB results. This would provide a visual that might be more meaningful than results in mol/L.

DISCUSSION.

Line 553. Should be more speculative here, particularly in light of the results.

Line 576. I’m not sure this serves as confirmation of the hypothesis (at least not particularly strong confirmation of the hypothesis).

CONCLUSION

Again, be more speculative related to the fNIR results and what they mean.

6. PLOS authors have the option to publish the peer review history of their article (what does this mean?). If published, this will include your full peer review and any attached files.

Reviewer #1: No

Reviewer #2: No

---

## [Author Response · Author response to Decision Letter 0]

4 May 2021

Dear Editor,

Dear Reviewers,

Thank you for the opportunity to revise our manuscript. Following the very useful comments and suggestions provided by both Reviewers, we have completed a careful revision our manuscript. We hope that you will find our revision satisfactory. We have also uploaded our responses as a separate document, in tabular format, which you might find easier to read. Please note that provided references (line numbers) to some of the changes made in the manuscript refer to the manuscript version with tracked changes.

Kind regards.

REVIEWER 1 

Comment 1. 

A cluster randomized controlled trial was conducted to investigate the effects of simple intervention, cognitively engaging active break intervention or control on children’s cognitive function and brain activity. Cognitively engaging active breaks produced a significant negative effect on sitting bouts greater than 5 minutes over mid and final time points and sit-to-stand transitions at mid-trial. The simple intervention and the cognitively engaging intervention showed a positive effect on sitting for 20 minutes or more, compared to the controls. 

Response to Comment 1.

We would like to thank the Reviewer for the useful comments and suggestions provided. We have addressed the comments to our best capacity, and hope that the manuscript has significantly improved following this revision.

Major revisions 

Comment 2. 

Table 5: Include time point in the model. If the interaction effect is significant, provide an interpretation of the results, but do not test main effects because the tests for main effects are uninteresting in light of significant interactions. If interaction effects are non-significant, drop the interaction effects from the model and test the main effects. Determining which results to present when testing interactions is often a multi-step process. We thank the Reviewer for this observation, which suggests that we can further improve the way we explain the type of analysis and factors we used and how they relate to the hypothesis testing. We have now incorporated a better explanation in the methods section and integrated the results to clarify the findings related to on-task behaviour.

Response to Comment 2.

A time*group interaction term is already incorporated into the analysis and tests whether there was a difference between groups over time. For this analysis, the dataset was placed in a long format and the variable “time” indicated the observation period (30-min pre and 30-min post). Hence, the ‘interaction term’ cannot be removed, as the test examines differences between groups over time.

To avoid confusion between the observation periods (30-min pre and 30-min post) and the trial timepoints (baseline, mid-trial, and end of trial) we have now ensured that observation periods are referred to as such throughout the manuscript.

Lines 487–491: “On-task behaviour data was summarised descriptively across the observation periods by group. We used multilevel generalized linear models to investigate the difference for children in the intervention groups in the odds of being observed on-task in the two consecutive 30-min periods compared to those in the control group, at each study time point”

Lines 700–724: “Table 5 presents the percentages of observation rated as on-task in each observation period divided by group. The results of the multilevel generalised linear mixed models (Table 6) show that, relative to the control group, the simple active breaks group had an 87% reduction in the odds of being observed on-task in the second 30-min observation period compared to the first (odds ratio [OR] = 0.13, 95% CI [0.05, 0.31], p = 0.001), at baseline. No statistical differences were noted between the control and the cognitively engaging group at this time point. At mid-trial, the simple active breaks group had 60% lower odds than the control of being observed on-task during the second in-class observation period compared to the first (OR = 0.40, 95% CI [0.18, 0.93], p < 0.05); the differences between the control and the cognitively engaging group during the two observation periods appeared not statistically significant at this time point. Although teachers were supposed to break up the lesson with an active break at this time point, the observers noted that the active breaks were not conducted between the two consecutive in-class observations. At the end of trial, the simple active breaks and the cognitively engaging group showed higher odds than the control (127% and 178%, respectively) of being observed on-task during the second in-class observation period, compared to the first (OR = 2.27, 95% CI [1.02, 5.08], p < 0.05 and OR = 2.78, 95% CI [1.16, 6.66], p < 0.05, respectively). This time, the active breaks were conducted between the two in-class observations as per intervention intention.

A new Table 5 was added. Table 6 is a simplification of the previously submitted Table 5.

Comment 3. 

State and justify the study’s target sample size with a pre-study statistical power calculation. The power calculation should include: (1) the estimated outcomes in each group; (2) the α (type I) error level; (3) the statistical power (or the β (type II) error level); (4) the target sample size and (5) for continuous outcomes, the standard deviation of the measurements. 

Response to Comment 3.

Thank you for your remark. As we reported in the trial registration (Australian New Zealand Clinical Trials Registry (registration number ACTRN12618002034213) and in the Trial Study Protocol submitted with the manuscript, the sample size determination was based on previous similar research (e.g., Egger et al. 2019) and a well-documented small to moderate effect size of physical activity on executive functions (e.g., Alvarez-Bueno et al., 2016; deGreeff et al., 2018); hence, an a-priori power calculation was not conducted. Nevertheless, we have now conducted a post-hoc power calculation to determine the attained power (1– β error probability), based on the actual sample size, the observed effect sizes of the intervention groups for each of the main outcomes, and α error probability = 0.05. We have also calculated the observed design effect (i.e., the correction in power required to account for the random effects of class as a clustering variable). All the above has now been reported in the Methods and Results sections; we have also discussed this aspect and acknowledged the limitation of our potential lack of power in the Discussion section.

Alvarez-Bueno, C., Pesce, C., Cavero-Redondo, I., Sanchez-Lopez, M., Pardo-Guijarro, M. J., & Martinez-Vizcaino, V. (2016). Association of physical activity with cognition, metacognition and academic performance in children and adolescents: a protocol for systematic review and meta-analysis. BMJ Open, 6(6), e011065. https://doi.org/10.1136/bmjopen-2016-011065

Egger, F., Benzing, V., Conzelmann, A., & Schmidt, M. (2019). Boost your brain, while having a break! The effects of long-term cognitively engaging physical activity breaks on children’s executive functions and academic achievement. PLoS ONE, 14(3), e0212482. https://doi.org/10.1371/journal.pone.0212482

de Greeff, J. W., Bosker, R. J., Oosterlaan, J., Visscher, C., & Hartman, E. (2018). Effects of physical activity on executive functions, attention and academic performance in preadolescent children: a meta-analysis. Journal of Science and Medicine in Sport, 21(5), 501–507. https://doi.org/10.1016/j.jsams.2017.09.595

Lines 216–219: “The sample size was determined based on previous research [e.g., 56] and a well-documented small to moderate effect size of physical activity on executive functions [e.g., 5,58]. We aimed to recruit around 43 children with typical development per study arm (⁓N = 130 children) and four teachers per study arm (⁓N = 12 teachers).”

Lines 456–467: “A measure of effect size (Cohen’s f2) was provided for fixed effects of the study groups, which was calculated according to the method described by Selya et al. [78]. This operation requires dividing the proportion of variance explained by the predictor of interest by the residual variance not explained by the model. Convetionally, the effects are considered small for f2 = 0.02, moderate for f2 = 0.15 and large for f2 = 0.35 [79]. Based on the observed effect size for each of the main outcomes, the actual sample size, and an α error probability = 0.05, a post-hoc power analysis was conducted to determine the attained power (1 – β error probability), also considering the design effect correction required to account for the random effects of class as a clustering variable. The design effect was calculated using the formula: 1 + [(CV2 + 1) × n – 1] × (ICC), where CV indicates the coefficient of variation for n; n is the number of students in each classroom (cluster); and ICC is the intraclass correlation coefficient from the linear mixed models.” 

Lines 609–616: “The observed effects of the study groups were negligible for working memory and lapses of attention (both f2 < 0.01), small for inhibition inverse efficiency score, response time and accuracy (f2 = 0.04, f2 = 0.02 and f2 = 0.02, respectively) and small to moderate for DL-PFC haemodynamic response (oxy: f2 = 0.05; deoxy: f2 = 0.07). A post-hoc analysis of the sample size revealed that the attained power was below the conventional 0.80 threshold for all main outcomes (1 – β < 0.55). The design effect did not influence most of the outcomes (< 2 conventionally indicated as the value below which the clustering in the data must not be accounted for), except for lapses of attention that showed a design effect of 2.73.” 

Lines 895–908: “The post-hoc power calculation based on the observed effects revealed that the sample size in our study was potentially small to allow us detecting significant differences. Despite the limited power, our study detected a significant effect of the cognitively engaging active breaks in the proportion of deoxygenated haemoglobin in the left dorsolateral prefrontal cortex. Thus, it provided useful information on the higher sensitivity of neural as compared to behavioural cognitive measures in this type of ecological school-based intervention. Also, the provided effect sizes are useful for estimating sample size to conduct future well-powered studies. It is worth noting that the observed effect sizes in the main outcome were smaller than the effects generally observed in previous studies [e.g., 56], presumably depending on implementation factors. It is possible that the breaks were too short or not sufficiently frequent/intense to generate the expected cognitive response in children. Indeed, the results from our mediation analysis showed that response inhibition was affected by the active breaks depending on the extent of change in sitting and standing time elicited by the breaks.”

Minor revisions 

Comment 4. 

Abstract: Indicate that the schools were randomly assigned to intervention. 

Response to Comment 4.

Thank you for your suggestion. This information is provided in Lines 13–15: “Classrooms from two consenting schools were randomly allocated to a six-week simple or cognitively engaging active break intervention. Classrooms from another school acted as a control group.” Please note that the control group could not be randomly allocated; we had already provided details on this aspect in the methods section (see Randomisation, Lines 243–250).

Comment 5 

Line 349: State the underlying covariance structure used in the linear mixed models and the criteria for selecting it. 

Response to Comment 5.

Thank you for this comment. The covariance structure used for the linear mixed models is reported in lines 449–455: “All models adjusted for sex and age and the baseline value of the outcome variable (to avoid regression to the mean [77]) and accounted for the random effects of classroom as a clustering variable. The models examining DL-PFC haemodynamic response were also adjusted for the performance change score in the cognitive task”

We have now added more details to clarify the criteria for selecting the confounders. Age and sex were selected because they are commonly reported as potential confounders in physical activity interventions. Specifically, physical activity has systematically been observed to decline with age and boys are often reported to be more active than girls. Differences in cognitive functions also appear to be influenced by these factors, i.e.: executive functions improve with age and previous research shows that biological sex influences the performance at some executive functioning tasks, generally favouring girls (e.g., Aadland et al., 2018). 

Since in the main analysis we used the change scores from baseline to end of trial as the outcome of each variable of interest, we also adjusted for the baseline value to avoid regression to the mean (Barnett et al., 2005). To account for the variability in haemodynamic response that could be explained by a change in cognitive performance, the models analysing DL-PFC haemodynamic response were also adjusted for the performance change score in the cognitive task.

Aadland, K., Aadland, E., Andersen, J. R., Lervåg, A., Moe, V., Resaland, G. K., & Ommundsen, Y. (2018). Executive Function, Behavioral Self-Regulation, and School Related Well-Being Did Not Mediate the Effect of School-Based Physical Activity on Academic Performance in Numeracy in 10-Year-Old Children. The Active Smarter Kids (ASK) Study. Frontiers in Psychology, 9. https://doi.org/10.3389/fpsyg.2018.00245

Barnett, A.G., van der Pols, J.C., Dobson, A.J. (2005). Regression to the mean: What it is and how to deal with it. Int J Epidemiol, 34: 215–220. doi:10.1093/ije/dyh299 

Lines 449–456: “All models were adjusted for sex and age—commonly identified as potential factors affecting cognitive functions, physical activity, and sedentary behaviour—and the baseline value of the outcome variable—to avoid regression to the mean [77]—and accounted for the random effects of classroom as a clustering variable. The models examining DL-PFC haemodynamic response were also adjusted for the performance change score in the cognitive task, to account for the variability in haemodynamic response that could be explained by a change in cognitive performance.”

Comment 6 

Line 527 and beyond: Clarify what summary statistic is represented by the letter M. 

Response to Comment 6.

Thank you for your suggestion. This is now clarified.

Line 728: “(simple intervention: mean [M] = 4.19, standard deviation [SD] =…”

Comment 7 

Table 1: In the statistical analysis section, include the statistical methods used to compare groups in table 1. 

Response to Comment 7.

We have now included this in the statistical analysis section. 

Lines 435–437: “Differences by study group in demographic characteristics (i.e., age, sex, reported medical/developmental condition, primary language spoken at home) were calculated using Analysis of Variance (ANOVA) or χ2 test according to the nature of the data.”

REVIEWER 2 

Comment 8. 

Breaking up classroom sitting time with cognitively engaging physical activity: Behavioral and brain responses. In this research, the authors examined a maximum of 141 children between the ages of 6 – 8 years old that participated in a simple or a cognitively engaging active break intervention for 6 weeks or were part of a control group. Baseline and post-intervention measurements were taken of response inhibition, attentional lapses, working memory, and relative changes in brain hemodynamic response to determine the effect of the interventions. Differences in HHB in the cognitively engaged group suggested increased brain efficiency in that group, although no groups differences existed in working memory or lapses of attention. Response inhibition improved in the cognitively engaging active breaks group.

With this study, the authors ask an important question: What are the impacts of different types of activity breaks on measures important to children’s success in the classroom? While it is tempting to believe that cognitively engaging active breaks would be of greater benefit to children, the topic itself has received very little empirical attention. The current study goes further and attempts to look at underlying mechanisms (or at least differences in brain oxygenation) that may change as a result of the different types of interventions. Overall, the study is ambitious and the design is appropriate. My biggest concerns are related to the fNIR data and the conclusions drawn from hemodynamic measures, particularly considering that the key conclusions the authors draw come from these data. The use of fNIR data to quantify group differences is innovative and interesting. At the same time, data collection, analysis, and interpretation can be fraught with error and therefore, requires documentation and detail to ensure the process was performed in a way that ensured the best data fidelity. 

Response to Comment 8.

We would like to thank the Reviewer for the careful analysis of our work and the useful comments and suggestions provided. We have addressed the Reviewer’s comments to best of our ability, and we believe the manuscript has improved significantly following this revision. Specific changes detailed in the following comments.

Comment 9. 

Overall, the methodology is lacking in specific details related to the use of the Artinis PortaLite system. Based on the description of the fNIR system, it provided hemodynamic information from a single channel fitted to the left prefrontal cortex for each child. 

Response to Comment 9.

Thank you for your important comment. We agree with the Reviewer that further details on this measure may be useful to enable the reader to better comprehend what we have done and provide a supporting base for the results we presented and their subsequent interpretation. To reduce the amount of information, we originally referred to one of our previous papers where we described the methodological procedures in greater detail, including the ones related to the use of fNIRS (Mazzoli et al., 2019; see Line 275–277). However, we understand your comment and consequently have now included further details on the fNIRS system in the Methods section, adding supporting information in a new supplementary table (S2 Table) that contains the data filtering parameters used for the data analysis in MATLAB. Also, as detailed in our response to Comment 34, we have expanded the background information in our Introduction section.

Mazzoli E, Teo WP, Salmon J, Pesce C, He J, Ben-Soussan TD, et al. Associations of class-time sitting, stepping and sit-to-stand transitions with cognitive functions and brain activity in children. Int J Environ Res Public Health. 2019/04/28. 2019;16: 1482. doi:10.3390/ijerph16091482

Lines 324–360: “Given the relative transparency of human tissues to near-infrared light, the emission of near-infrared light into a tissue and the measurement of the intensity of re-emerging light enables assessment of changes in O2Hb and HHb non-invasively in that specific region [17]. Specific details on the technological and methodological aspects related to fNIRS is available several previous reviews [e.g., 66]. Each child involved in this assessment had the fNIRS probe fitted by the corresponding author (EM) on their forehead in the area corresponding to the left DL-PFC. From the frontal aspect of forehead, the landmark corresponding to the left DL-PFC was identified as the area between the mid-point of the left eyebrow and the hair growth. The 10–10 international system [67] was used to position the probe’s light source and detector (in AP3 and F5, respectively), in line with the approach described by Zimeo Morais et al. [68]. The probe was secured in place using a dark elastic fabric hairband. The probe size is 58 × 28 × 6 mm, and it fits quite precisely on the small forehead of a child. No skin preparation was required but the researcher ensured that no hair was in the way and that the incoming signal was good prior to starting the assessment. An illustration of the probe and the experimental setup is available in Fig 2. 

For each child, the real-time changes in O2Hb and HHb in the left DL-PFC were recorded while performing a 10-min task involving inhibition abilities (Go/No-Go task). Collected data were processed, and artefacts were removed, using the hemodynamic optically measured evoked response (HomER2), a MATLAB-based (MATLAB R2017a, MathWorks, Natick, MA, USA) user interface developed by Huppert and colleagues [69] and artefacts were removed as described by Brigadoi et al. [70]. The artefacts removal process involved the following steps: i) raw data were converted in optical density changes; ii) a filtering algorithm was applied to detect motion artefacts; iii) a principal component analysis (PCA) was applied to correct the detected motion artefacts; iv) an high pass filter (0.010 Hz) and a low pass filter (0.20 Hz) were applied to clear the data from the high and low frequency noise; and v) optical density data were converted in concentration changes and used for the analysis. A detailed description of the utilised pipeline values for data processing, also including the MATLAB script, is available as supporting information (S2 Table).”

Further details in Fig 2 and S2 Table. For the changes applied to the introduction please refer to our response to Comment 24.

Comment 10. 

Who placed the sensor and how was the location above the left prefrontal cortex verified? With EEG, a 10 – 20 placement system is frequently used to standardize the location of electrode placement; how did the authors ensure standardized placement, particularly considering the smaller size of children’s foreheads? 

Response to Comment 10.

Thank you for this important point. EEG and fNIRS have several differences, but they generally share the same approach when it comes to the placement of electrodes/optodes. The 10-20 method, that the Reviewer referred to, has been traditionally used to distribute electrodes/optodes with precision and reproducibility. Given that data collection was conducted in the school premises (not in a laboratory), we opted for a single channel fNIRS device (Portalite by Artinis) for its portability and quick and easy setup. To ensure that the probe placement was consistent, the fNIRS experiment was conducted by the same researcher (first author, EM) throughout the trial. The probe size is 58 × 28 × 6 mm and it fits quite precisely on the small forehead of a child, in the area between the eyebrow and the scalp with hair growth. From the mid-point of forehead (aligned with the nasion) the researcher identified the area corresponding to left dorsolateral prefrontal cortex (probe’s light source and detector in AP3 and F5, respectively) using the mid-point of the left eyebrow and the hair growth as points of reference to triangulate the optimal landmark for the optode positioning. The optode was secured in place using a dark elastic fabric hairband. A graphical representation of the setup and experiment has now been included (Fig 2).

Lines 331–340: “From the frontal aspect of forehead, the landmark corresponding to the left DL-PFC was identified as the area between the mid-point of the left eyebrow and the hair growth. The 10–10 international system [67] was used to position the probe’s light source and detector (in AP3 and F5, respectively), in line with the approach described by Zimeo Morais et al. [68]. The probe was secured in place using a dark elastic fabric hairband. The probe size is 58 × 28 × 6 mm, and it fits quite precisely on the small forehead of a child. No skin preparation was required but the researcher ensured that no hair was in the way and that the incoming signal was good prior to starting the assessment. An illustration of the probe and the experimental setup is available in Fig 2.”

Comment 11. 

Was the skin prepared prior to placement? How was the signal optimized (e.g., noise reduction) for each participant, given that individual differences exist in skin transparency based on skin color? 

Response to Comment 11.

No preparation is needed with fNIRS. However, we made sure that the probe was in touch with the skin and no hair was in the way. Since skin colour is known to affect the quality of the signal, we checked that the incoming signal was good prior to starting the test; the proprietary software (Oxysoft) provides information on the quality of incoming light and no issues were identified by the software during the preparation stage. 

Lines 337–339: “No skin preparation was required but the researcher ensured that no hair was in the way and that the incoming signal was good prior to starting the assessment.”

Comment 12. 

The authors should provide this and other information related to the fNIR system, particularly because many readers interested in this research may not be familiar with the technology. 

Response to Comment 12.

We have now included further information on fNIRS and referred to additional references on the technological and methodological aspects of this approach. Please refer to our response to Comment 9 and Comment 24.

Comment 13. 

Another issue comes from the interpretation of the fNIR data. The authors collected data from the left PFC, which is a relatively small brain region involved in multiple cognitive activities. At the same time, any individual cognitive activity consists of a functional network of interacting brain structures, so there is not necessarily a clear correlation between a ‘greater positive change in the proportion of HHB’ that the cognitively active children demonstrated and an increase in cognitive efficiency. Also, in many studies, O2Hb is the more sensitive biomarker; the authors may want to discuss why that measure does not replicate the HHB results (and could potentially describe the differences in the two measures). 

Response to Comment 13.

We agree regarding the complexity of the interacting brain areas that underlie the expression of different cognitive abilities. However, we rely on evidence that the association between neural activity in the dorsolateral prefrontal cortex and executive functions is well documented (Diamond, 2002; Fiske, 2019; Miller, 2009). The typical response observable in response to neural activity is a positive change in O2Hb and a negative change in HHb, which is nicely summarised by Scholkmann et al. (2014, Fig. 5). During our assessment, children were performing a cognitive task involving inhibition processes (Go/No-Go task). The pre- to post- average changes in O2Hb and HHb were used for the analyses and models were adjusted for the cognitive performance at the Go/No-go and other confounders (i.e., baseline O2Hb/HHb, sex, age, class [clustering variable]). This allowed us to determine whether changes O2Hb and HHb in the left dorsolateral prefrontal cortex could be dependent on the study group, amongst other factors. We have now modified the relevant sentence in the Discussion section, with further explanations provided below.

As the Reviewer rightfully pointed out, the O2Hb signal is typically more sensitive than HHb (Scholkmann, 2014). However, the HHb signal was found to have better spatial specificity compared to O2Hb (Dravida et al. 2018). We are unsure about whether this can provide a definite answer, but we believe it may partially explain why we found an effect with HHb and not with O2Hb. In relation to the differences between signals, we have already attempted to provide an explanation in our Discussion section: “Patterns of O2Hb and HHb often appear in alternation, meaning that an increase in O2Hb is accompanied by a reduction in HHb and vice versa [66]. Also, the algorithms used to calculate O2Hb and HHb are based on different wavelengths (usually 760 and 850 nm, respectively) [66].” (Lines 774–777). 

Furthermore, the predicted margins for the model including O2Hb as an outcome shows that changes were somewhat reflective of what we observed with HHb (i.e., the cognitively engaging group had a negative change in O2Hb while the control group had a significant positive change in O2Hb while controlling for cognitive performance at the cognitive task). This was already reported in Lines 781–788, now slightly reworded to further clarify it. 

Diamond, A. (2002). Normal development of prefrontal cortex from birth to young adulthood: Cognitive functions, anatomy, and biochemistry. Principles of Frontal Lobe Function, 466–503.

Dravida, S., Noah, J. A., Zhang, X., & Hirsch, J. (2018). Comparison of oxyhemoglobin and deoxyhemoglobin signal reliability with and without global mean removal for digit manipulation motor tasks. Neurophotonics, 5(1), 11006. https://doi.org/10.1117/1.NPh.5.1.011006

Fiske, A., & Holmboe, K. (2019). Neural substrates of early executive function development. Developmental Review, 52, 42–62. https://doi.org/https://doi.org/10.1016/j.dr.2019.100866

Miller E K and Wallis J D (2009) Executive Function and Higher-Order Cognition: Definition and Neural Substrates. In: Squire LR (ed.) Encyclopedia of Neuroscience, volume 4, pp. 99-104. Oxford: Academic Press

Scholkmann, F., Kleiser, S., Metz, A. J., Zimmermann, R., Mata Pavia, J., Wolf, U., & Wolf, M. (2014). A review on continuous wave functional near-infrared spectroscopy and imaging instrumentation and methodology. NeuroImage, 85, 6–27. https://doi.org/10.1016/j.neuroimage.2013.05.004

Lines 776–788: “Also, the algorithms used to calculate O2Hb and HHb are based on different wavelengths (usually 760 and 850 nm, respectively) [66]. The resulting signals in response to neural activity typically show more sensitive changes in O2Hb compared to HHb [66]. However, the HHb signal was found to have better spatial specificity compared to O2Hb [89], which might partially explain our significant findings in relation to the HHb but not O2Hb. Despite the absence of statistical significance, the results of the mixed models show that both active conditions, compared to the control, had negative average changes in O2Hb (although not statistically significant) and positive average changes in HHb (only significant for the cognitively engaging group) while controlling for the cognitive performance at the concurrent cognitive test, which aligns with our hypothesis of improved DL-PFC efficiency for children in the cognitively engaging intervention.”

Comment 14. 

In that regard, the fNIR results should be more speculative, and perhaps be a secondary focus of the study rather than a primary focus. 

Response to Comment 14.

We would like to thank the reviewer for this suggestion. We have changed the discussion and conclusions which referred to haemodynamic response to present a more cautious interpretation of the findings. However, we believe this aspect should still be an important aim, especially considering that the sample size is large and the study design novel for an fNIRS study. Indeed, our primary aim was to test the effects of active breaks on cognitive functions (including the underlying haemodynamic changes that may explain differences between groups). 

Lines 757–759: “This is suggestive of more efficient neural activity in the DL-PFC due to the lower level of metabolic resources (haemodynamic response) necessary to achieve the same level of cognitive performance [66].”

Lines 892–894: “It is possible that the study was not sufficiently powered to detect significant differences in all outcomes of interest, particularly for the cognitive tests that involved a reduced sample (i.e., working memory and fNIRS).”

Lines 921–923: “Another limitation is the employment of a single channel fNIRS device, which has limited spatial resolution compared to a whole-head system.” 

Lines 925–927: “Our study adds to the literature with suggestive evidence supporting the notion that cognitively engaging active breaks may help to improve children’s neural activity efficiency in the DL-PFC (neural substrate of executive functions)”

Comment 15. 

Below are specific questions or comments within each section of the manuscript.

Response to Comment 15.

Thank you for the detailed suggestions.

ABSTRACT 

Comment 16. 

Line 5 and 8. The authors refer to ‘brain activity’ in general. They should be more specific here to the prefrontal cortex, since that is the only measure of brain activity (more specifically, brain oxygenation), that they are observing. 

Response to Comment 16.

We have changed the terminology as appropriate throughout the manuscript: ‘brain activity’ was replaced with ‘dorsolateral prefrontal cortex (DL-PFC) haemodynamic response’ throughout the manuscript.

Comment 17. 

Line 11. Age range of participants in abstract (6 – 8 years) is different than in text (6 – 9 years, on page 8). 

Response to Comment 17.

Thank you for your attentive reading. However, kindly note that this is not an error. The age range of consenting participants reported in the abstract is correct and so is the sentence on Line 210 which refers to the inclusion criteria for eligibility to participate, rather than the characteristics of those who agreed to participate, as we stated: “Typically developing children aged between 6 and 9 years, both males and females, attending Grade 1 and 2 in a mainstream primary school were eligible to participate.”

We have now modified this sentence to clarify that this is referred to the eligibility criteria. Line 210: “To be eligible to participate in the study children had to be i) typically developing ii) aged between 6 and 9 years, and iii) attending Grade 1 and 2 in a mainstream primary school.”

Comment 18. 

Line 11. The reported sample size is misleading, especially considering that for working memory and fNIR assessments, the authors used about half that number. Provide a little more detail on participant number per analysis somewhere in the methods. 

Response to Comment 18.

We understand that the different number of children completing different assessments can be confusing. However, the sentence in the abstract was not meant to be misleading but only appropriate for the brevity requirement of this section and it did specify that not all children completed all assessments: “Up to 141 children, aged between 6 and 8 years (46% girls), were included, although not all of them completed each assessment.” We have now modified this sentence to be clearer.

A clear breakdown of those numbers was already provided in the manuscript (Lines 220–230 and in Fig 1). The reason for this difference is related to the time required to conduct working memory and fNIRS assessments, particularly considering that we conducted research in the real school environment, not in a lab. We have already specified this reason in Line 226 and in the legend of Fig 1. 

Lines 11–13: “Up to 141 children, aged between 6 and 8 years (46% girls), were included, although about half of them completed two of the assessments (n = 77, working memory; n = 67, dorsolateral prefrontal cortex haemodynamic response).”

Comment 19. 

Line 17. It is probably more appropriate to use the term “haemodynamic response” rather than “brain activity” 

Response to Comment 19.

We have applied the suggested change. Line 18: “event-related brain haemodynamic response (dorsolateral prefrontal cortex).”

Comment 20. 

Line 29. I recommend using the word “suggestive” rather than “indicative” to be more speculative.

Response to Comment 20.

We have applied the suggested change. Lines 31–32: “…which under the same cognitive performance is suggestive of improved neural efficiency.”

Comment 21 

Line 37. This conclusion seems appropriately speculative. 

Response to Comment 21.

Thanks for this positive comment.

Comment 22 

Line 45, # 1. I would argue that the authors have not proved this point and should not list it as their first key point. 

Response to Comment 22.

We agree with the reviewer that the highlight provided as #1 was not reflective of our study results. We have now modified it to align it with what we found. 

Lines 49–53: “1. Children doing cognitively engaging active breaks showed a positive change in the proportion of deoxygenated haemoglobin in the left dorsolateral prefrontal cortex (neural substrate of executive functions) compared to the control group, while controlling for cognitive performance, which is suggestive of improved efficiency in that brain region.” 

INTRODUCTION 

Comment 23.

Overall, the introduction is well written, but needs more information on Haemodynamics of the brain and what that means. 

Response to Comment 23.

We have included significant relevant additions to the introduction section as we detailed in response to Comment 24. 

Comment 24. 

Line 123. Somewhere in this section, the authors should expand their discussion of neural measures and haemodynamic response. For example, there is no mention of O2Hb or HHB (what they measure, how they change with neural activity, what they might mean). The authors could also introduce other research that uses these measures as an indication of cognitive efficiency. 

Response to Comment 24.

We have included more detailed introduction regarding fNIRS. Please note that there are multiple technical and methodological aspects pertaining to fNIRS and they cannot all be presented in this manuscript. These aspects, however, were reported in detail in several reviews that have now been included as appropriate citations in our Introduction and Methods sections. 

Lines 88–113: “Cognitive functions are traditionally assessed by analysing the behavioural responses to a computer-based test that supposedly challenges certain functions, and the performance is generally quantified in terms of response time and accuracy rate. The concurrent use of objective measures of brain activity might provide complementary evidence of the underlying mechanisms that support changes in cognitive performance, although such approaches have rarely been applied to physical activity interventions. The most advanced method to investigate changes in brain structure and function with high spatial resolution involves the use of neuroimaging techniques, such as functional magnetic resonance imaging. A recent systematic review identified nine studies that employed neuroimaging techniques in youth to test the effects of physical activity interventions, with findings from seven randomised controlled trials included in the review showing significant improvements in brain structure and/or function [15]. Despite these promising findings, most neuroimaging devices are non-portable and high cost, which may explain the paucity of physical activity research using this technique. Another approach is to measure event-related brain potentials using electroencephalography, which—although has high temporal resolution and is significantly cheaper than most neuroimaging techniques—is not easily portable, has low spatial resolution, and is very sensitive to motion artefacts. Functional near-infrared spectroscopy (fNIRS) is a relatively novel optical technique that allows to measure brain changes of oxygenated (O2Hb) and deoxygenated (HHb) haemoglobin (i.e., haemodynamic response) [16]. Its high spatial and temporal resolution, high biochemical specificity, the portability, and the relative stability to motion artefacts make fNIRS greatly advantageous in terms of ecological validity compared to other techniques [17]. A fast-growing number of studies, mostly cross-sectional in design, have employed fNIRS in youth [18]. To the authors’ knowledge, only two cross-sectional studies [19,20] and no interventions pertaining to children’s physical activity and brain function have been conducted using fNIRS.”

For the changes applied to the Methods section refer to Comment 9, 10 and 31.

Comment 25. 

Line 131. State how long the intervention took place. 

Response to Comment 25.

This information was already provided in the abstract (line 14), in Fig 1, and in the Methods section (line 256–257: “…twice a day (between 9:00 am and 11:00 am and between 11:30 am and 1:00 pm) for six weeks”). We have now also included this detail in the paragraph describing the aim of the study at the end of the Introduction, as suggested, and we have stated it again in the methods section.

Lines 162–163: “the effects of a six-week active break intervention”

Lines 272–273: “The trial was carried out for six weeks, between October and December 2017.”

Comment 26. 

Line 132. What does “normal practice” mean? 

Response to Comment 26.

The normal school practice involves no breaks apart from recess and lunch. Line 166: “(normal school practice with no breaks apart from recess and lunch)”

Comment 27 

Line 142. “higher” efficiency, not “better” efficiency. 

Response to Comment 27.

This was changed. Line 178: “…would reflect improved efficiency in the active conditions compared to the control”

METHOD 

Comment 28.

Line 169. Ages are different than reported in Abstract. 

Response to Comment 28.

Please see our response to Comment 17.

Comment 29. 

Line 222. Did the authors check for fidelity of teachers presenting intervention? 

Response to Comment 29.

Teacher’s fidelity was assessed using activity logs. The results were generally indicative that the intervention was implemented quite regularly. Two teachers did not return the activity log they were asked to complete but reported that the activities were performed as prescribed. We also used the change values in sitting, standing stepping patterns across class time and school time to check for fidelity. However, we could not check how well teachers delivered those activities. We have now provided additional information on these aspects.

Lines 426–430: “The differences by study group on children’s sitting, standing, and moving patterns over time during class/school periods were used to assess teacher’s adherence to the program. Additional information was retrieved from teachers’ activity logs, which were designed to allow teachers to record the number and type of active breaks performed on each trial day.”

Lines 469–471: “To test fidelity of the intervention, linear mixed models examined the effects of the active breaks on the change values in sitting, standing, and stepping patterns across class time and school time. Additionally, teachers’ activity logs were inspected.”

Lines 735–740: “Activity logs were collected from teachers and suggested that most activities were implemented as prescribed. However, the analysis of children’s sedentary patterns suggested that significant changes were only observed by the end of trial and not at mid-trial. It is possible that this finding is reflective of the time teachers need to familiarise with new activities before implementing them regularly.”

Comment 30. 

Line 264, Is the Working Memory validated for 6 years old? 

Response to Comment 30.

Yes, Tulsky et al. (2013) produced evidence reliability and validity for children aged between 3 and 15 years. We have corrected lines 312–314 to reflect this. 

Lines 312–314: “…an iPad-based cognitive test which demonstrated excellent test-retest reliability and adequate convergent and discriminant validity in children and adolescents (3–15 years) [61]”

Comment 31. 

Line 270 – 284. Expand this section (see above). It may be useful to include a figure depicting a participant with the Artinis system on. 

Response to Comment 31.

We have now included an illustration (Fig 2) providing details on the fNIRS system used, the placement on the forehead and the experimental setting during the data collection regarding haemodynamic response. 

Comment 32. 

Line 354. Were all the variables normally distributed? If not, were any adjustments made? 

Response to Comment 32.

The variables were normally distributed and we have now specified this in the results. Line 558: “Outcome variables appeared approximately normally distributed”

RESULTS 

Comment 33. 

Overall, the results were presented in a way that is easy to follow and understand. 

Response to Comment 33.

Thank you for this positive comment. 

Comment 34. 

Line 449 – 456. The authors might consider using graphs to present the O2Hb and HHB results. This would provide a visual that might be more meaningful than results in mol/L. 

Response to Comment 34.

Thank you for the useful suggestion. We have added graphical representation to present the raw O2Hb and HHb responses together with the adjusted concurrent cognitive performance, at end of trail (Fig 3).

Also, we have now included a graphical presentation of the results for all the main outcomes for i) all participants and ii) excluding participants with a reported medical or developmental condition (Fig 4). 

DISCUSSION

Comment 35. 

Line 553. Should be more speculative here, particularly in light of the results. 

Response to Comment 35.

We agree and have thus made our interpretation more speculative. Line 757–759: “This is suggestive of more efficient neural activity in the DL-PFC due to the lower level of metabolic resources (haemodynamic response) necessary to achieve the same level of cognitive performance [66].” 

Comment 36. 

Line 576. I’m not sure this serves as confirmation of the hypothesis (at least not particularly strong confirmation of the hypothesis). 

Response to Comment 36.

We have mitigated our statement. Line 786–788: “which aligns with our hypothesis of improved DL-PFC efficiency for children in the cognitively engaging intervention.”

CONCLUSION 

Comment 37. 

Again, be more speculative related to the fNIR results and what they mean. 

Response to Comment 37.

This was changed to be more speculative. Lines 925–927: “Our study adds to the literature with suggestive evidence supporting the notion that cognitively engaging active breaks may help to improve children’s neural activity efficiency in the DL-PFC (neural substrate of executive functions) and improve response inhibition abilities.”

---

## [Decision Letter · Decision Letter 1]

14 Jun 2021

Breaking up classroom sitting time with cognitively engaging physical activity: behavioural and brain responses

PONE-D-20-22144R1

Dear Dr. Mazzoli,

We’re pleased to inform you that your manuscript has been judged scientifically suitable for publication and will be formally accepted for publication once it meets all outstanding technical requirements.

Kind regards,

Maciej S. Buchowski

Academic Editor

PLOS ONE

Additional Editor Comments (optional):

Reviewers' comments:

Reviewer's Responses to Questions

**Comments to the Author**

1. If the authors have adequately addressed your comments raised in a previous round of review and you feel that this manuscript is now acceptable for publication, you may indicate that here to bypass the “Comments to the Author” section, enter your conflict of interest statement in the “Confidential to Editor” section, and submit your "Accept" recommendation.

Reviewer #1: All comments have been addressed

Reviewer #2: All comments have been addressed

2. Is the manuscript technically sound, and do the data support the conclusions?

Reviewer #1: (No Response)

Reviewer #2: Yes

3. Has the statistical analysis been performed appropriately and rigorously? 

Reviewer #1: (No Response)

Reviewer #2: Yes

4. Have the authors made all data underlying the findings in their manuscript fully available?

Reviewer #1: (No Response)

Reviewer #2: Yes

5. Is the manuscript presented in an intelligible fashion and written in standard English?

Reviewer #1: (No Response)

Reviewer #2: Yes

6. Review Comments to the Author

Reviewer #1: (No Response)

Reviewer #2: (No Response)

7. PLOS authors have the option to publish the peer review history of their article (what does this mean?). If published, this will include your full peer review and any attached files.

Reviewer #1: No

Reviewer #2: No

---

## [Editor Report · Acceptance letter]

18 Jun 2021

PONE-D-20-22144R1 

Breaking up classroom sitting time with cognitively engaging physical activity: behavioural and brain responses 

Dear Dr. Mazzoli:

I'm pleased to inform you that your manuscript has been deemed suitable for publication in PLOS ONE. Congratulations! Your manuscript is now with our production department. 

Kind regards, 

on behalf of

Dr. Maciej S. Buchowski 

Academic Editor

PLOS ONE